# Maintenance of proteostasis by *Drosophila* Rer1 is essential for competitive cell survival and Myc-driven overgrowth

Pranab Kumar Paul[1,⦿], Shruti Umarvaish[1,⦿], Shivani Bajaj[1], Rishana Farin S.[1], Hrudya Mohan[1], Wim Annaert[2], Varun Chaudhary[1]*

1 Cell and developmental signaling laboratory, Department of Biological Sciences, Indian Institute of Science Education and Research Bhopal, Bhopal, Madhya Pradesh, India, 2 Laboratory for Membrane Trafficking, VIB-Center for Brain and Disease Research, KU Leuven, Leuven, Belgium, and Department of Neurosciences, KU Leuven, Gasthuisberg, Leuven, Belgium

⦿ These authors contributed equally to this work.
* varun.c@iiserb.ac.in

**Data Availability Statement:** All relevant data are within the paper and its Supporting Information files.

## Abstract

Defects in protein homeostasis can induce proteotoxic stress, affecting cellular fitness and, consequently, overall tissue health. In various growing tissues, cell competition based mechanisms facilitate detection and elimination of these compromised, often referred to as 'loser', cells by the healthier neighbors. The precise connection between proteotoxic stress and competitive cell survival remains largely elusive. Here, we reveal the function of an endoplasmic reticulum (ER) and Golgi localized protein Rer1 in the regulation of protein homeostasis in the developing *Drosophila* wing epithelium. Our results show that loss of Rer1 leads to proteotoxic stress and PERK-mediated phosphorylation of eukaryotic initiation factor 2α. Clonal analysis showed that *rer1* mutant cells are identified as losers and eliminated through cell competition. Interestingly, we find that Rer1 levels are upregulated upon Myc-overexpression that causes overgrowth, albeit under high proteotoxic stress. Our results suggest that increased levels of Rer1 provide cytoprotection to Myc-overexpressing cells by alleviating the proteotoxic stress and thereby supporting Myc-driven overgrowth. In summary, these observations demonstrate that Rer1 acts as a novel regulator of proteostasis in *Drosophila* and reveal its role in competitive cell survival.

## Author summary

In developing tissues, cells can stochastically acquire defects that can reduce their fitness. To maintain the overall health of tissues, these unfit cells are identified by the healthier neighboring cells and eliminated via a juxtacrine-acting cellular fitness sensing mechanism called cell competition. An example of such physiological regulation of cellular fitness is the maintenance of proteostasis. Defects in maintaining proteostasis cause proteotoxic stress. Interestingly, proteotoxic stress is observed not only in the unfit loser cells but also in the overgrowing super-competitor cells, for instance, cells with higher levels of Myc. How cell competition is linked to the maintenance of proteostasis is poorly

**Funding:** This work was supported by the Science and Engineering Research Board (SERB), Department of Science & Technology, Government of India (grant number: CRG/2021/004686 to VC). The laboratory of V.C. is also supported by intramural funds from IISER Bhopal and the Department of Biotechnology-EMR (grant number: BT/PR34467/BRB/10/1831/2019 to VC). P.K.P. received fellowship from the Council of Scientific & Industrial Research (09/1020/(0127)/2017-EMR-I). W.A. acknowledges the financial support of the Vlaams Instituut voor Biotechnologie (VIB), KU Leuven (grant number: C14/21/095 and KA.20/085 to WA), the Fonds Wetenschappelijk Onderzoek (FWO) (grant number: I001322N to WA), and the Stichting Alzheimer Onderzoek België (grant number: #2020/0030 to WA). The funders had no role in the study design, data collection and analysis, decision to publish, or preparation of the manuscript.

**Competing interests:** The authors have declared that no competing interests exist.

understood. In this study, we have characterized for the first time the function of *Drosophila* Rer1 protein in development. We demonstrate that Rer1 is essential for maintaining protein homeostasis and loss of Rer1 activates stress-induced unfolded protein responses. Cells lacking Rer1 are identified as unfit cells and become losers when juxtaposed to the normal neighboring cells. Moreover, we show that Myc-overexpressing cells upregulate Rer1 levels, which allows them to maintain a higher demand for stress regulation, caused by increased protein translation. In this work, we propose that Rer1 functions as a stress regulator and that modulating its levels could provide cytoprotection under stress conditions.

## Introduction

The development of healthy tissue requires the removal of viable but suboptimal cells. In several growing tissues, this vital culling process is orchestrated through a specific cell-cell interaction called cell competition. In this intricate mechanism, unfit cells, also called "loser", are eliminated by their surrounding fitter counterparts, the "winner" cells [1,2], thereby maintaining tissue health [3]. The best-known example of cell competition is described in the developing *Drosophila* epithelium using the heterozygous mutations in a ribosomal protein (*Rp*) gene (also known as *Minute*). The $Rp^{+/-}$ flies are viable, however, under mosaic condition the $Rp^{+/-}$ cells are eliminated from the developing epithelium when juxtaposed with the neighboring wild-type ($Rp^{+/+}$) cells [4–6]. Although the $Rp^{+/-}$ mutation affects cellular physiology autonomously, caspase-dependent apoptosis is observed mostly at the boundary between $Rp^{+/-}$ cells and nearby $Rp^{+/+}$ cells, which is a hallmark of cell competition [6,7]. The loser fate of the slow growing $Rp^{+/-}$ cells was suggested to be due to reduced protein translation [8–10]. However, recent studies have shown that $Rp^{+/-}$ cells exhibit high proteotoxic stress [11–14] and activate the expression of bZip transcription factor Xrp-1, which plays an essential role in the elimination of the $Rp^{+/-}$ cells [11,15,16]. Interestingly, Xrp-1 appears to be responsible for the manifestation of various defects in $Rp^{+/-}$ cells, including reduced global translation and proteotoxic stress, contributing to the loser status [17].

Moreover, the loser fate is associated with a number of other physiological changes impacting cell fitness. These changes include, 1) reduced metabolic activity due to alteration in the mTOR pathway activity [18,19], 2) loss of apico-basal polarity as a consequence of mutations of the *scribble*, *dlg*, and *lgl* genes [20], 3) defects in endosomal trafficking caused by mutations in the *rab5* gene [21], and 4) deregulation of signaling pathways such as Wnt, BMP, and Hippo [22–24]. Cells bearing these perturbations are eliminated through cell competition involving JNK-dependent activation of the proapoptotic factors [6,25].

Interestingly, some perturbations can also provide a competitive advantage to the cells over their wild-type neighbors. For instance, the overexpression of a proto-oncogene *Myc*, a master regulator of cell proliferation and growth, enhances the relative fitness of the cells. Thus, clonal expression of Myc generates super-competitor cells, which proliferate at the expense of the wild-type neighbors [10,26]. Myc drives cellular growth through its ability to upregulate the expression of a large number of genes and enhance the activity of several crucial metabolic pathways [27–29]. However, Myc-overexpression also leads to proteotoxic stress due to increased protein synthesis [30–32]. Thus, Myc-driven overgrowth is dependent on the activation of the cytoprotective unfolded protein response pathways (UPR) [33]. This includes phosphorylation of eukaryotic initiation factor 2 alpha (eIF2α) via PERK (PKR-like ER kinase) and induction of autophagy to reduce protein translation and clear misfolded proteins, respectively

[34,35]. However, a clear understanding of how Myc and UPR cooperate to promote a proliferative cellular environment remains unclear.

Here, we investigated the role of Retention in Endoplasmic Reticulum-1 (Rer1) protein in the competitive cell proliferation in the developing *Drosophila* wing epithelium. Mutations in the *rer1* gene were first described in yeast, where it was identified in a screen as a factor required for proper transport of Sec12p between the endoplasmic reticulum (ER) and Golgi [36]. Later studies have shown that Rer1 is also required for the assembly of multisubunit protein complexes, for example, the tetrameric γ-secretase complex, yeast iron transporter and skeletal muscle nicotinic acetylcholine receptor (nAChR) [37–42]. Rer1 is also known to regulate ER homeostasis, and therefore loss of Rer1 has been shown to induce ER stress in yeast and worms [43]. Despite the fact that Rer1 is evolutionarily conserved from yeast to mammals, its function in the development of organisms remains largely unknown [43–45].

By creating a *rer1* loss-of-function mutant, we show that *rer1* is an essential gene in *Drosophila*. Furthermore, we found that loss of Rer1 creates proteotoxic stress in the developing wing epithelium, and when surrounded by wild-type cells, the clonal population of *rer1* mutant cells attained the loser fate and were eliminated specifically via the process of cell competition. We have also analyzed the role of Rer1 in Myc-induced overgrowth and Rer1 levels were found to be upregulated upon Myc-overexpression. More importantly, we found that loss of Rer1 is sufficient to suppress Myc-induced overgrowth. In summary, our results demonstrate that Rer1 is an essential protein for proper maintenance of protein homeostasis and competitive cell survival in a developing tissue.

## Results

### Rer1 is required for *Drosophila* larval development

We first set out to characterize the role of Rer1 during *Drosophila* development. To this end, we generated a *rer1* knockout mutant by imprecise excision of a p-element insertion in the *rer1* locus (see materials and methods). A loss-of-function mutation in *rer1* containing a 1560 bp deletion in the coding region was identified (**Fig 1A**). Quantitative RT-PCR analysis in the homozygous mutant ($rer1^{-/-}$) animals confirmed a complete loss of *rer1* mRNA levels, indicating a complete loss-of-function (**S1A Fig**). Further analysis showed that the $rer1^{-/-}$ larvae failed to develop into pupae and died during the larval stages (**S1B Fig**). To rule out the possibility of lethality arising due to a second site mutation in another essential gene, we performed rescue experiments using a genomic-rescue construct expressing GFP-tagged Rer1 via the endogenous promoter (see materials and methods). The expression of *GFP-rer1* in homozygous $rer1^{-/-}$ flies led to a complete rescue of lethality, confirming the specificity of the mutant (**S1B Fig**). These results underscore the indispensability of Rer1 in *Drosophila* development.

### Cells lacking Rer1 show reduced survival in the developing wing epithelium

We next analyzed the importance of Rer1 at the tissue level using the developing *Drosophila* wing imaginal discs. We first depleted Rer1 in the posterior compartment of the wing discs by expressing *rer1*-RNAi using the *hedgehog (hh)-Gal4* driver (**S1C Fig**). To test the efficiency of the knockdown, we expressed *rer1*-RNAi in the *GFP-rer1* genomic-rescue flies. Here, we observed a strong downregulation of the GFP-Rer1 levels (**S1D Fig**), suggesting that *rer1*-RNAi effectively downregulated the Rer1 levels. We assessed the impact of Rer1 depletion on cell death by analyzing the levels of cleaved Death caspase-1 (Dcp-1) and Acridine Orange (AO) as apoptosis markers. Rer1 depletion in the posterior compartment led to a strong increase in both Dcp-1 and AO positive cells as compared to the control anterior compartment (**S1E–S1H Fig;** quantified in **S1I and S1J Fig**, respectively). Intriguingly, despite the increased

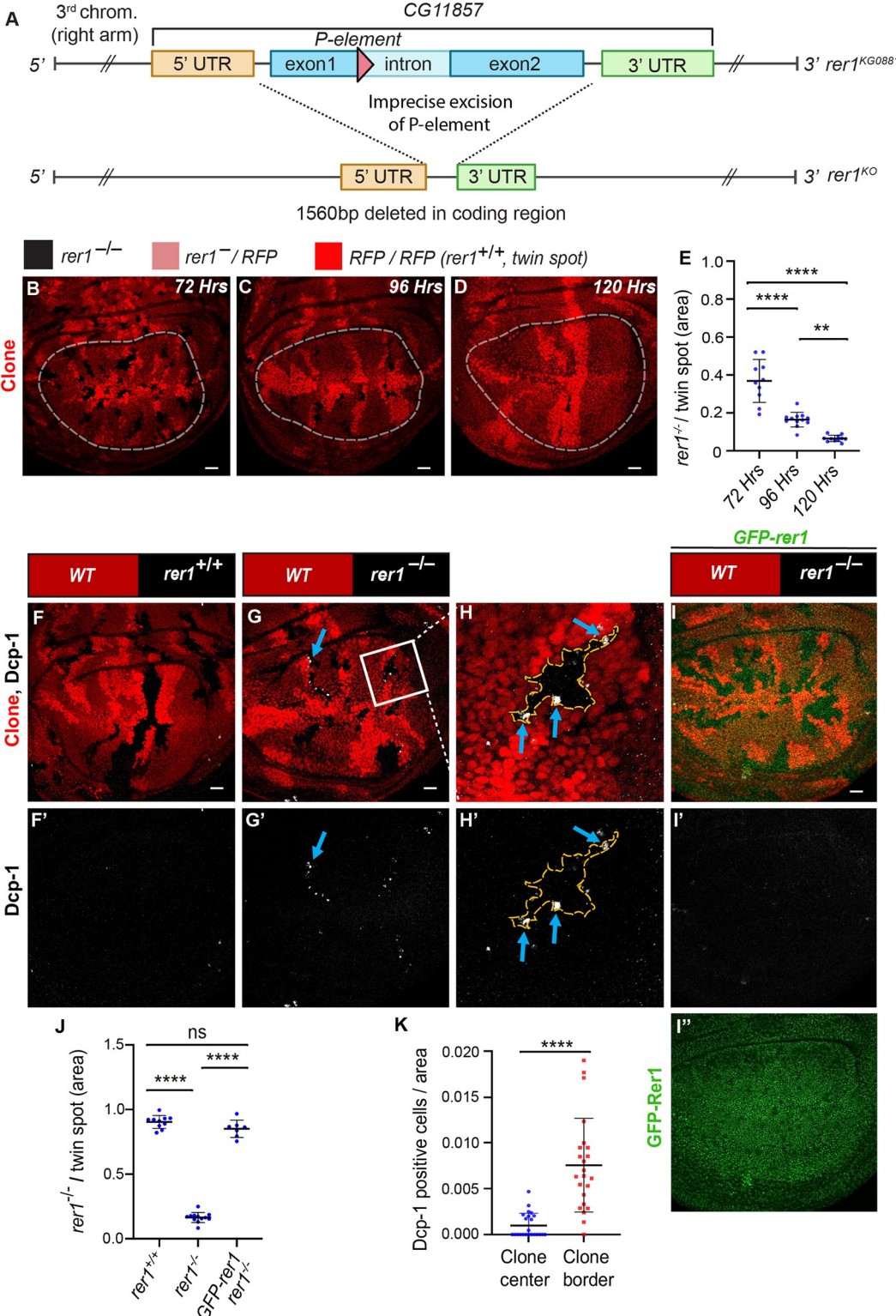

**Fig 1. *rer1⁻/⁻* clones show reduced growth and cell death at the clone boundary. (A)** Schematic representation of the *rer1^{KO}* line. Upon imprecise excision of a p-element inserted in the coding sequence, a 1560bp deletion in the *rer1* gene was obtained. **(B-D)** Wing imaginal disc harboring *rer1⁻/⁻* clones induced by hs-FLP at 72, 96 and 120 hrs prior to dissection of third-instar larvae. RFP-negative (black) represents *rer1⁻/⁻*, lighter red areas represent heterozygous *rer1⁻/RFP*, and brighter red areas represent RFP/RFP (*rer1^{+/+}*; twin spot). **(E)** The relative size of mutant (RFP-negative) versus twin spots (RFP/RFP)

areas at 72 hrs (N = 10 wing discs), 96 hrs (N = 12 wing discs), and 120 hrs (N = 11 wing discs), measured within the white dotted lines. Statistical analysis was performed using the Ordinary one-way ANOVA with Tukey's multiple comparison test (**** p<0.0001, ** p<0.0036). **(F-I)** Third-instar larval wing epithelium with hs-FLP-induced (96 hrs AHS) mitotic clones of **(F-F')** WT (wild-type; *rer1*$^{+/+}$), and **(G-G')** *rer1*$^{-/-}$ genotypes, immuno-stained for the anti-cleaved Dcp-1. **(H-H')** A magnified image of the inset (white box) in **G**. **(I-I")** *rer1*$^{-/-}$ clones in GFP-rer1 background stained with anti-cleaved Dcp-1. **I"** shows the expression of GFP-Rer1 in the wing imaginal disc. **(J)** Quantification of the relative size of *rer1*$^{-/-}$(RFP-negative) versus twin spots (RFP/RFP) areas in WT control (**F**, N = 10 wing discs); *rer1*$^{-/-}$(**G**, N = 12 wing discs) and rescue in GFP-Rer1, *rer1*$^{-/-}$(**I**, N = 6 wing discs). Statistical analysis was performed using the Ordinary one-way ANOVA with Tukey's multiple comparison test (**** p<0.0001). **(K)** Quantification of cell death at the center and border of *rer1*$^{-/-}$clones (two-sided Wilcoxon signed-rank test; N = 23 clones present in 12 wing discs; **** p<0.0001, SB = 20 μm. Also see **S1** and **S2** **Figs**.

cell death, the adult wing of these flies appeared normal (**S1K–S1N Fig**; quantified in **S1O Fig**).

To delve further, we generated *rer1*$^{-/-}$clones using the heat-shock-inducible Flippase (FLP)-Flp recognition target (FRT)-system (see materials and methods). Clones were induced during early larval stages (48 hrs AEL) and wing discs were dissected at 72 and 96 hrs after heat-shock (AHS). Moreover, some larvae that were delayed and could reach up to 120 hrs were also dissected and analyzed. In these experiments, we observe that the *rer1*$^{-/-}$(RFP-negative) clones area reduced over time as compared to the *rer1*$^{+/+}$ clones (RFP/RFP; also called twin spot) (**Fig 1B–1D**; quantified in **Fig 1E**), indicating progressive removal of *rer1*$^{-/-}$cells from the epithelium. Moreover, generation of the *rer1*$^{-/-}$clones did not alter the overall wing size (**S2A–S2D Fig**; quantified in **S2E Fig**), indicating that the loss of *rer1*$^{-/-}$cells was compensated by the neighboring cells.

## *rer1*$^{-/-}$cells are eliminated through cell competition

To validate these results, we analyzed the Dcp-1 levels in *rer1*$^{-/-}$clones. Consistent with the RNAi experiments, *rer1*$^{-/-}$clones showed upregulation of Dcp-1 levels (**Fig 1F–1H**), which was rescued by the expression of GFP-Rer1 (**Fig 1I–1I"**). Additionally, the *rer1*$^{-/-}$clone growth was rescued by the expression of GFP-Rer1 (**Fig 1J**; quantification of RFP-negative area in **Fig 1F, 1G** and **1I**). Notably, Dcp-1 positive cells were concentrated at the boundary of *rer1*$^{-/-}$cells and neighboring control cells (**Fig 1H–1H'** see blue-arrows; quantified in **Fig 1K**; also see **S2F–S2G Fig**), indicative of elimination via cell competition, a phenomenon observed in *Minute/+* cells when competing with normal cells [6,7,19,46–49].

While these results suggest that cell death in *rer1*$^{-/-}$clones could arise from competition between two different population of cells, the occurrence of cell death upon depletion of Rer1 in the entire posterior compartment (**S1E–S1H Fig**), a non-competitive setting, required further analysis. Thus, we revisited the impact of Rer1 depletion on cell death by generating *rer1*-RNAi expressing MARCM clones (**S3A–S3B Fig**). Remarkably, these clones exhibit higher cell death at the clone boundary (**S3B'–S3B" Fig** see blue arrows; quantified in **S3C Fig**) and reduction in the clone size as compared to control (**S3D Fig**), consistent with our observation in the *rer1* mutant clones, indicating that loss of Rer1 could trigger cell competition.

Next, we sought to further confirm that the elimination of *rer1*$^{-/-}$cells occurs via cell competition. A hallmark of cell competition is that the loser or winner fate of the cells depends upon the relative fitness with the neighboring cells [50,51]. Thus, we asked if the fate of *rer1*$^{-/-}$cells could be altered by reducing the fitness of their neighbors. To this end, we selected *ribosomal protein S3* (*RpS3*) mutant, which also creates loser cells. However, the homozygous *RpS3* mutant (*RpS3*$^{-/-}$) cells show autonomous cell lethality, while the heterozygous *RpS3* mutant (*RpS3*$^{+/-}$) cells are eliminated by surrounding wild-type cells (*RpS3*$^{+/+}$) via cell competition [6]. Utilizing this paradigm, we generated *rer1*$^{-/-}$clones in *RpS3* heterozygous mutant background, causing juxtaposition of *rer1*$^{-/-}$cells with the *RpS3*$^{+/-}$cells. Growth of wild-type cells in the *RpS3* heterozygous background was used as a control. Here, we observed a dramatic

increase in the growth of $rer1^{-/-}$ clones, although not to the same extent as control cells versus $RpS3^{+/-}$ (**Fig 2A–2D**; quantified in **Fig 2E**). More importantly, the Dcp-1 staining could now be observed in the $RpS3^{+/-}$ cells at the boundary (**Fig 2D'–2D"** see blue arrows; quantified in **Fig 2F**), demonstrating that the activation of cell death in $rer1^{-/-}$ could be reversed by reducing the fitness level of the neighbors (**Fig 2G–2H**). Thus, the boundary cell death in $rer1^{-/-}$ clones observed in otherwise normal background is due to higher fitness of the surrounding wild-type cells. Altogether, these results show that the loss of Rer1 created loser cells which are eliminated via cell competition.

## Loss of Rer1 creates proteotoxic and oxidative stress

Studies have shown that Rer1 is localized dynamically between the ER and cis-Golgi compartments in *Saccharomyces cerevisiae* [52,53] and mammalian cells [40,44], where it functions in protein quality control processes [45]. However, its localization and function in *Drosophila* remains unknown. Utilizing the *GFP-rer1* genomic rescue construct, we observed that GFP-Rer1 colocalized with both ER (Calnx) and Golgi (Golgin-245) markers in the wing epithelial cells (**S4A–S4B Fig**), affirming its conserved localization.

Next, we wondered if the loss of Rer1 also affected protein homeostasis in flies. Thus, we analyzed the level of phosphorylated eIF2α (p-eIF2α), which is a well-established marker of proteotoxic stress [54]. Consistent with the proposed function of Rer1 [43], higher levels of p-eIF2α were observed in both $rer1^{-/-}$ clones (**Fig 3A–3C**) and RNAi-mediated depletion of Rer1 (**S5A–S5F Fig**) compared to the control regions, indicating activation of the UPR pathways. The increased level of p-eIF2α in $rer1^{-/-}$ clones was restored upon the expression of *GFP-rer1* (**Fig 3D–3F**).

We further investigated if the loss of Rer1 also affected redox homeostasis, which has been linked to the activation of cell death response post ER stress [55–57]. Thus, we analyzed the production of reactive oxygen species (ROS) upon loss of Rer1, via Dihydroethidium (DHE) labeling (S1 Text). We observed that the expression of *rer1*-RNAi in the posterior compartment led to an increase in the levels of DHE as compared to the control anterior compartment, indicating high oxidative stress (**S6A–S6D Fig**). Altogether, these results suggest that loss of Rer1 increases both proteotoxic and oxidative stress in cells.

## PERK-mediated phosphorylation of eIF2α causes elimination of *rer1* mutant cells

Four kinases, namely, PERK, GCN2 (general control nonderepressible 2), PKR (protein kinase R) and HRI (heme regulated inhibitor) are known to sense cellular stress and cause phosphorylation of eIF2α, however, in *Drosophila* only PERK and GCN2 are conserved [58]. Studies suggest that phosphorylation of eIF2α via PERK is due to ER stress [59,60], whereas the phosphorylation via GCN2 is due to amino acid starvation [61,62]. Thus, to further dissect the mechanisms of eIF2α phosphorylation upon loss of Rer1, we depleted PERK or GCN2 in $rer1^{-/-}$ cells, using the MARCM approach (see material and methods), and analyzed p-eIF2α levels. We observed that knockdown of PERK in $rer1^{-/-}$ clones reduced the levels of p-eIF2α, while GCN2 depletion did not show any significant change (**Fig 4A–4D,** compare **Fig 4B''', 4C''' and 4D''';** quantified in **Fig 4E**). Consistent with this, we observed that the expression of *PERK*-RNAi in the posterior compartment caused reduction in the p-eIF2α, in both control and *rer1*-RNAi expressing discs (**S7A–S7B and S7D–S7E Fig**). Whereas coexpression of *GCN2*-RNAi and *rer1*-RNAi did not alter p-eIF2α levels (**S7F Fig**), although expression of *GCN2*-RNAi alone showed mild reduction (**S7C Fig**). Altogether, these results show that loss of Rer1 caused ER stress leading to PERK-mediated phosphorylation of e-IF2α.

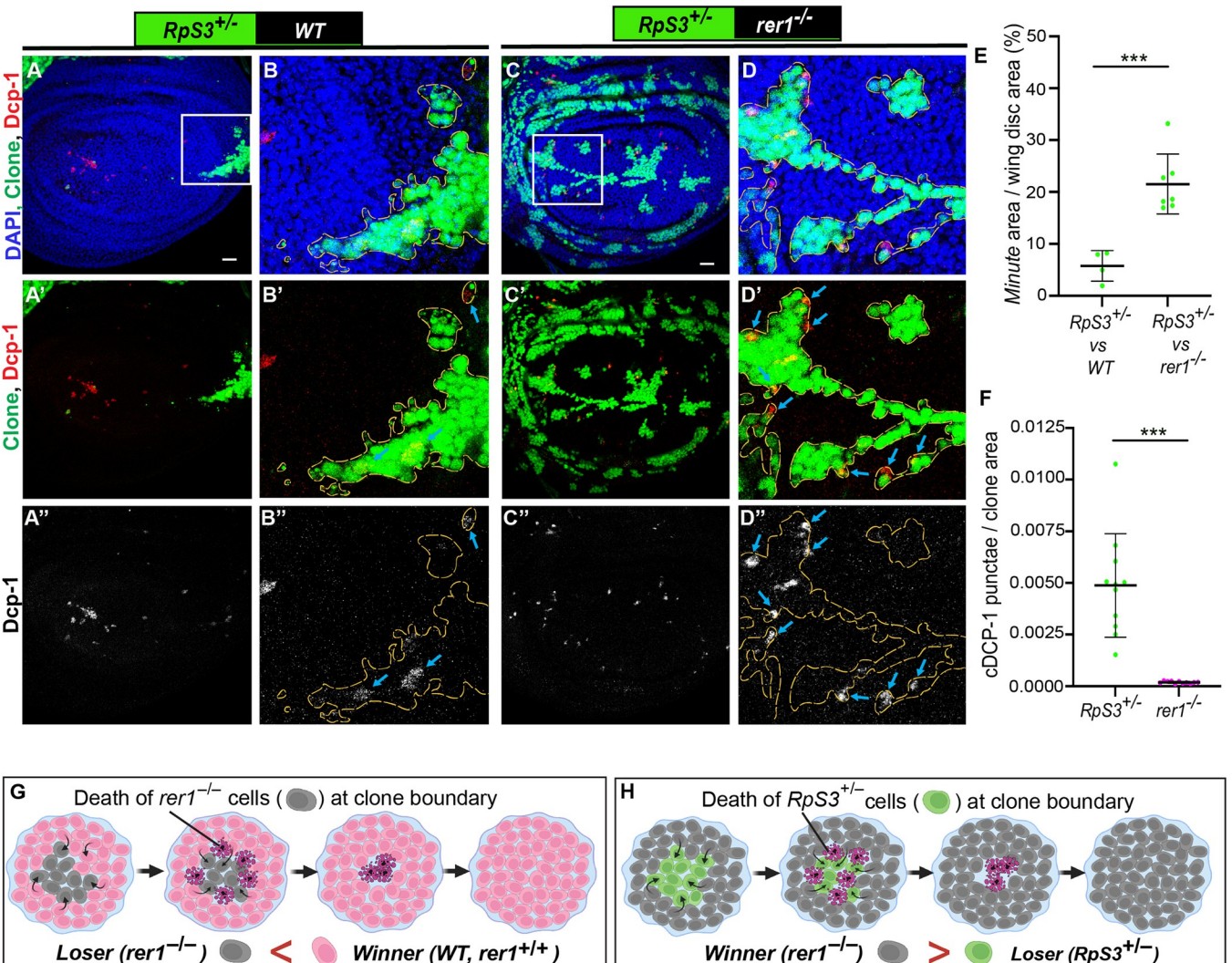

**Fig 2. *rer1⁻/⁻* cells are eliminated through cell-competition. (A-D)** Representative images of *hs*-FLP-induced (96 hrs AHS) mosaic wing imaginal discs containing heterozygous *RpS3⁺/⁻* cells (GFP-positive) juxtaposed to either **(A-B)** wild-type (WT) cells (GFP-negative) or **(C-D)** *rer1⁻/⁻* cells (GFP-negative), stained with the anti-cleaved Dcp-1. **(E)** Quantification shows percentage coverage of the GFP positive *Minute* area in *RpS3⁺/⁻ vs WT (A, N = 4 wing discs) or RpS3⁺/⁻ vs rer1⁻/⁻* (**C**, *N = 7 wing discs*). Statistical analysis in **E** was performed using the two-tailed Welch's t-test (***p = 0.0002). **(F)** Quantification of Dcp-1 positive cells in *RpS3⁺/⁻ vs rer1⁻/⁻* discs shows relatively higher levels of Dcp-1 positive cells in the GFP-positive *RpS3⁺/⁻* tissue as compared to the GFP-negative *rer1⁻/⁻* region (Wilcoxon paired t test; N = 12 wing discs; *** p = 0.0005). **(G-H)** Schematic diagram illustrating the concept of winner and loser fate in cell competition between WT (*rer1⁺/⁺*) and *rer1⁻/⁻* tissues; and *rer1⁻/⁻* and *RpS3⁺/⁻* tissues, respectively. SB = 20 μm.

We also noticed that depletion of PERK, but not GCN2, significantly increased the size of the *rer1⁻/⁻* clones (quantified in **Fig 4F**), indicating that higher levels of p-eIF2α may have a negative effect on the survival of *rer1⁻/⁻* cells. To ascertain whether PERK or GCN2 depletion also affected cell death in *rer1⁻/⁻* clones, we analyzed the Dcp-1 levels. We found that PERK depleted *rer1⁻/⁻* clones showed a reduction in Dcp-1, as compared to either control or *GCN2-RNAi* (**Fig 4G–4J**; quantified in **Fig 4K**), suggesting that the activation of PERK is responsible for the elimination of *rer1⁻/⁻* cells.

To further confirm these results, we tested the effect of dephosphorylation of p-eIF2α in *rer1⁻/⁻* cells. Thus, we used the overexpression of growth arrest and DNA damage-inducible 34 protein (GADD34), which provides specificity to protein phosphatase 1 for the dephosphorylation of p-eIF2α [63]. As expected, the overexpression of GADD34 in *rer1⁻/⁻* cells led to a

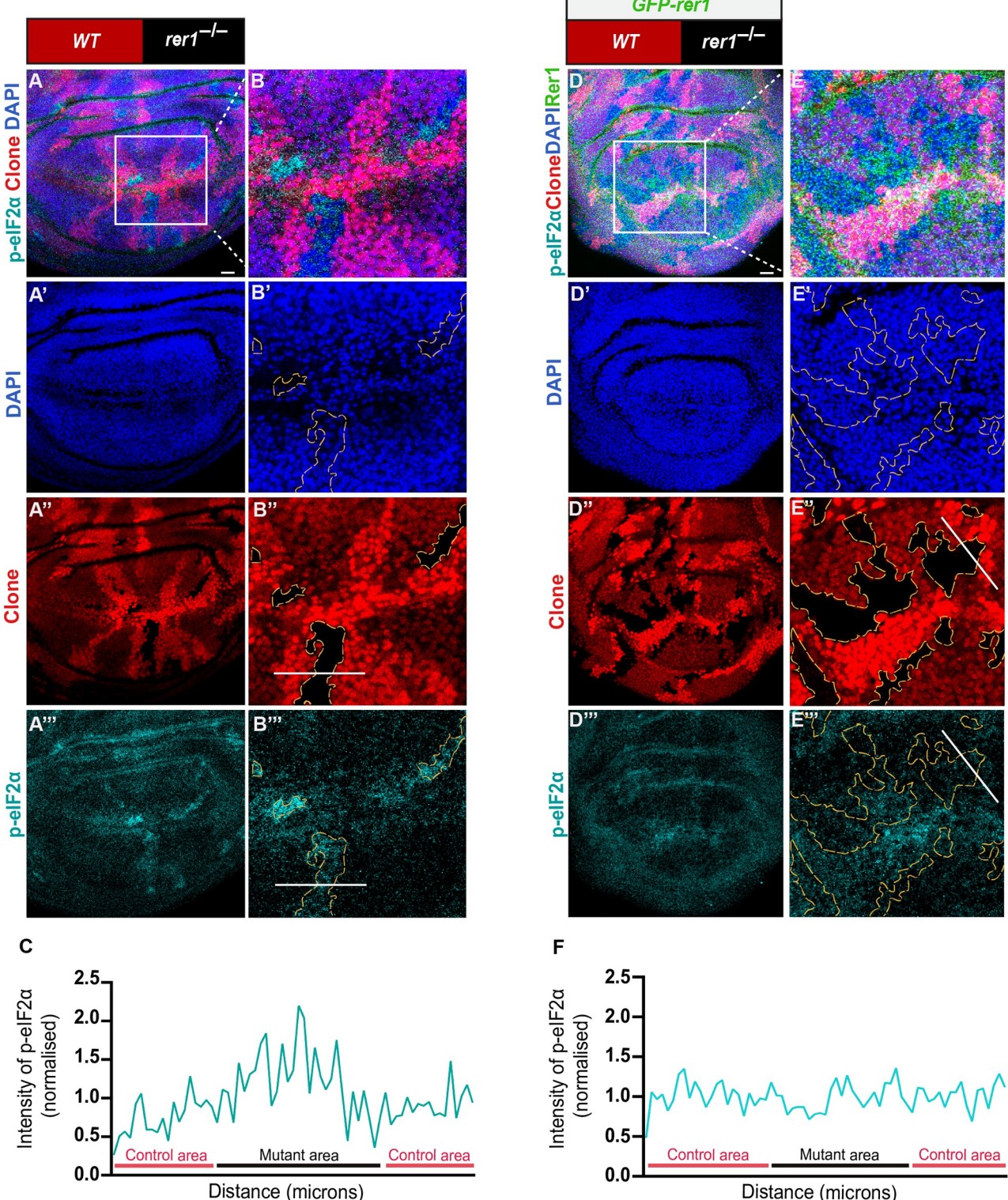

**Fig 3. Loss of *rer1* creates proteotoxic stress in the wing epithelial cells. (A)** hs-FLP-induced (96 hrs AHS) mitotic clones of *rer1⁻/⁻* tissues in third-instar larval wing epithelium, immuno-stained for the anti-p-eIF2α. **(B)** A magnified image of the inset (white box) in **A**. **(C)** Graph showing the intensity profile of p-eIF2α along the line ROI (white) in **B"** (N = 12 wing discs). **(D)** *rer1⁻/⁻* clones in GFP-rer1 background stained with anti-p-eIF2α. **(E)** A magnified image of the inset (white box) in **D**. **(F)** Graph showing the intensity profile of p-eIF2α along the line ROI (white) in **E'''** (N = 5 wing discs). SB = 20 μm. Also see **S4** and **S5 Figs**.

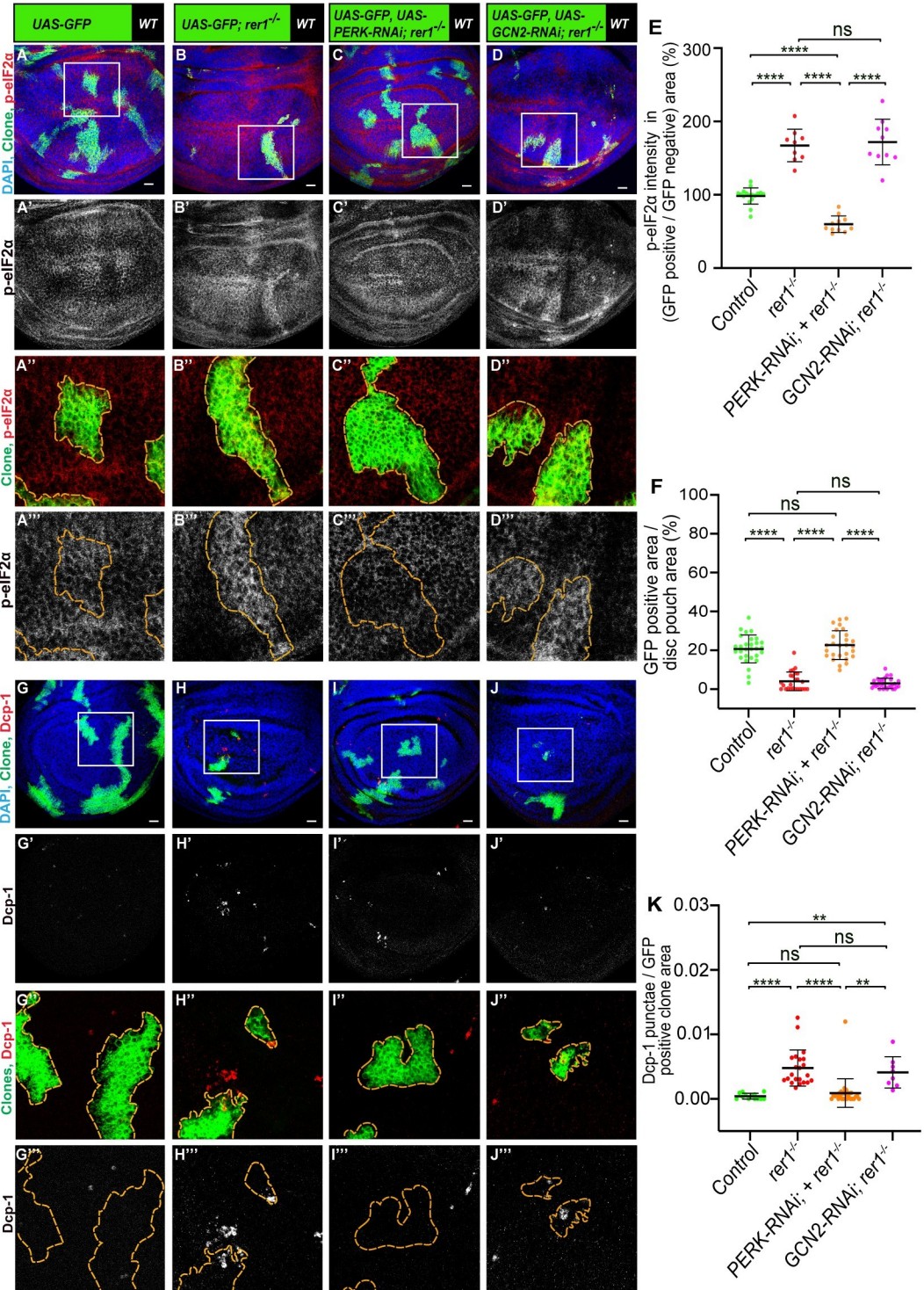

**Fig 4. PERK-mediated phosphorylation of eIF2α causes elimination of *rer1* mutant cells. (A-D)** Third-instar wing epithelium containing hs-FLP-induced MARCM clones (72 hrs AHS), of following genotypes, **(A)** *UAS-GFP (rer1⁺/⁺)*, **(B)** *UAS-GFP, rer1⁻/⁻*, **(C)** *UAS-GFP, rer1⁻/⁻ + UAS-PERK RNAi* and **(D)** *UAS-GFP, rer1⁻/⁻ + UAS-GCN2 RNAi*, stained with the anti-p-eIF2α antibody. **(E)** Quantification of the p-eIF2α levels inside the GFP-positive clones with respect to the nearby GFP-negative control tissue in, **A** (N = 17 wing discs), **B** (N = 9 wing discs), **C** (N = 11 wing discs) and **D** (N = 10 wing discs). Statistical analysis was performed using the Ordinary one-way ANOVA with Tukey's multiple comparison test (**** p<0.0001). **(F)** Quantification of the relative size of GFP-labeled clones area in; **A** (N = 29 wing discs), **B**, (N = 25 wing discs), **C** (N = 27 wing discs) and **D** (N = 29 wing discs). Statistical analysis was performed using the Ordinary one-way

ANOVA with Tukey's multiple comparison test (**** p<0.0001). **(G-J)** Third-instar wing epithelium containing hs-FLP-induced MARCM clones (72 hrs AHS) of following genotypes, **(G)** *UAS-GFP (rer1$^{+/+}$)*, **(H)** *UAS-GFP, rer1$^{-/-}$*, **(I)** *UAS-GFP, rer1$^{-/-}$ + UAS-PERK RNAi* and **(J)** *UAS-GFP, rer1$^{-/-}$ + UAS-GCN2 RNAi*, stained with the anti-Dcp-1 antibody. **(K)** Quantification of the Dcp-1 in the GFP-labeled clones area in, **G** (N = 13 clones in 3 wing discs), **H** (N = 22 clones in 9 wing discs), **I** (N = 28 clones in 8 wing discs) and **J** (N = 8 clones in 6 wing discs). Statistical analysis was performed using the Ordinary one-way ANOVA with Tukey's multiple comparison test (**** p<0.0001, ** p = 0.0029 between G and J, ** p = 0.0041 between I and J). SB = 20 μm. Also see **S7 Fig**.

strong reduction in the p-eIF2α levels (**Fig 5A–5B**; quantified in **Fig 5C**). Importantly, consistent with the effect of *PERK*-RNAi, the growth of *rer1$^{-/-}$* clones was rescued by the overexpression of GADD34 along with a concomitant decrease in Dcp-1 levels (**Fig 5D–5E**; quantified in **Fig 5F and 5G**, respectively). Similarly, coexpression of *rer1*-RNAi and GADD34 in the posterior compartment showed strong downregulation of p-eIF2α (**S8A Fig**) and Dcp-1 levels (**S8B Fig**). These results show that p-eIF2α played a maladaptive role promoting the elimination of *rer1$^{-/-}$* loser cells, unlike in the case of *Rp* mutants, where it was suggested to be adaptive [12] (see discussion).

Being an essential component of the translational machinery, phosphorylation of eIF2α is suggested to cause a reduction in global protein synthesis [54,59,60]. We sought to test whether higher levels of p-eIF2α in *rer1$^{-/-}$* also affects protein translation, which perhaps led to their elimination. Thus, we performed an O-propargyl-puromycin (OPP) incorporation assay, where OPP, an analog of puromycin, is incorporated into nascent polypeptide chains directly allowing rapid assessment of global protein synthesis [64]. Surprisingly, in these experiments, *rer1$^{-/-}$* cells did not show significant changes in the OPP levels (**S9A–S9B Fig**; quantified in **S9C Fig**). While these results are consistent with other reports where phosphorylation of eIF2α did not lead to a reduction in the global protein translation [65–67], it is also conceivable that *rer1$^{-/-}$* only have a mild effect on protein translation, that is undetectable with the OPP assay. Furthermore, the loser status of the *rer1$^{-/-}$* cells with almost normal translation is consistent with similar observations for other cell competition factors in recent studies [11,16].

## Cells lacking Rer1 activates JNK pathway that partially limits their growth

JNK signaling is intricately associated with multiple cellular stress pathways [25] and it is also involved in the elimination of loser cells through cell competition [6,57]. Given that the *rer1$^{-/-}$* cells are eliminated due to stress, we aimed to investigate the involvement of the JNK pathway in this process. We first tested if loss of Rer1 led to the activation of JNK signaling. The expression of a well-established JNK reporter, TRE-DsRed [68] was found to be upregulated in *rer1$^{-/-}$* clones (**Fig 6A–6C**). Additionally, the expression of two other known targets of JNK signaling, *puckered (puc) and Hid* [69–71] were also upregulated upon RNAi-mediated depletion of Rer1 and in *rer1$^{-/-}$* clones, respectively (**S10A–S10D Fig**). Collectively these results show that loss of Rer1 activates JNK signaling.

Subsequently, we explored whether the activation of JNK activity contributed to the elimination of the *rer1$^{-/-}$* cell. We observed that the expression of dominant-negative Basket (*bsk$^{DN}$*), the *Drosophila* ortholog of JNK [72], modestly improved the growth of *rer1$^{-/-}$* cells (**Fig 6D–6F**; quantified in **Fig 6G**), however, Dcp-1 levels remained unaffected (**Fig 6D–6F**; quantified in **Fig 6H**). This aligns with a previous report on the role of JNK in the survival of *Rps3$^{+/-}$* cells, where inhibition of JNK signaling rescued growth without altering cell death [57]. Nevertheless, we tested if the expression of *bsk$^{DN}$* in *rer1$^{-/-}$* clones altered the developmental progression of the larvae, which may have resulted in clone size difference. We find that the pupariation time for the larvae with *rer1$^{-/-}$* was similar to the larvae with *rer1$^{-/-}$* clones expressing *bsk$^{DN}$*, indicating that the expression of *bsk$^{DN}$* did not significantly change developmental

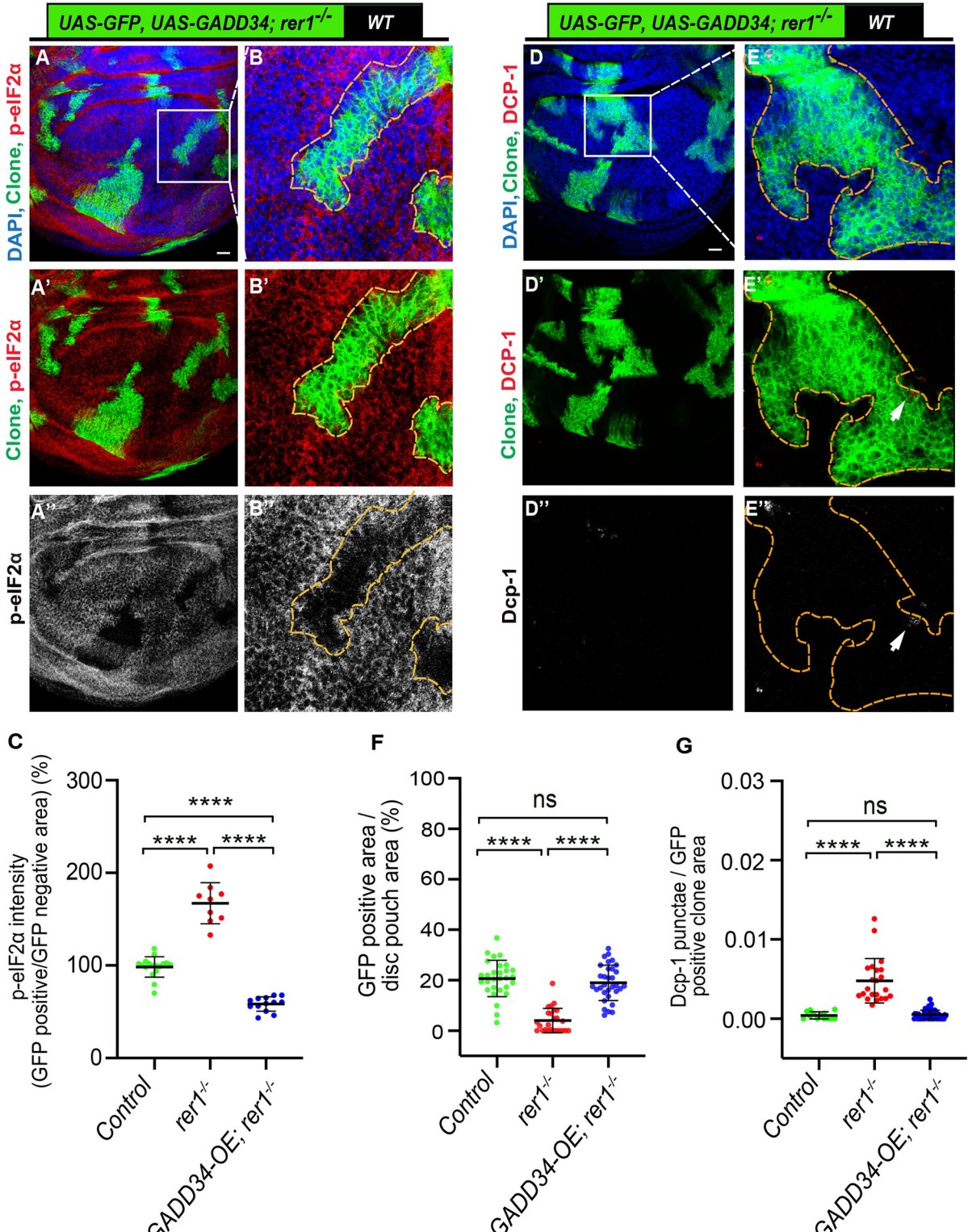

**Fig 5. Dephosphorylation of p-eIF2α rescues growth of *rer1* mutant cells. (A-B)** Representative images of the third-instar wing epithelium containing hs-FLP-induced MARCM clones (72 hrs AHS) of *UAS-GFP, UAS-GADD34; rer1⁻/⁻* genotype, stained with the anti-p-eIF2α antibody. **(B-B")** Magnified images of the insets in **A**. **(C)** Quantification of the p-eIF2α levels inside the GFP-positive clones with respect to the nearby GFP-negative tissue in control discs (*UAS-GFP*, same as in Fig 4A; N = 17 wing discs), *rer1⁻/⁻*(*UAS-GFP; rer1⁻/⁻*, same as in Fig 4B; N = 9 wing discs) and *GADD34-OE; rer1⁻/⁻*(*UAS-GFP, UAS-GADD34; rer1⁻/⁻*; N = 13 wing discs). Statistical analysis was performed

using the Ordinary one-way ANOVA with Tukey's multiple comparison test (**** p<0.0001). **(D-E)** Representative images of the third-instar wing epithelium containing hs-FLP-induced MARCM clones (72 hrs AHS) of *UAS-GFP, UAS-GADD34; rer1$^{-/-}$* genotype, stained with the anti-Dcp-1 antibody. **(E-E")** Magnified images of the insets in D. **(F)** Quantification of GFP-positive clone area in control discs (*UAS-GFP*, same as in Fig 4A; *N* = 29 wing discs), *rer1$^{-/-}$*(*UAS-GFP; rer1$^{-/-}$*same as in Fig 4B; N = 25 wing discs) and *GADD34-OE; rer1$^{-/-}$*(*UAS-GFP, UAS-GADD34; rer1$^{-/-}$*; N = 32 wing discs). Statistical analysis was performed using the Ordinary one-way ANOVA with Tukey's multiple comparison test (**** p<0.0001). **(G)** Quantification of Dcp-1 levels with respect to the GFP-positive clone area in control discs (*UAS-GFP*, same as in Fig 4G; N = 13 clones in 3 wing discs), *rer1$^{-/-}$*(*UAS-GFP; rer1$^{-/-}$*, same as in Fig 4H; N = 22 clones in 9 wing discs) and *GADD34-OE; rer1$^{-/-}$*(*UAS-GFP, UAS-GADD34; rer1$^{-/-}$*; N = 56 clones in 15 wing discs). Statistical analysis was performed using the Ordinary one-way ANOVA with Tukey's multiple comparison test (**** p<0.0001). SB = 20 μm. Also see S8 Fig.

timing of larvae harboring *rer1$^{-/-}$*clones (**S10G Fig**). Moreover, p-eIF2α levels remained high in *rer1$^{-/-}$*cells expressing *bsk$^{DN}$* (**S10E–S10F Fig**), placing JNK activity either downstream or parallel to stress induction. Altogether, these results suggest that activation of JNK signaling contributes towards the maladaptive effects of p-eIF2α in *rer1$^{-/-}$*cells, restricting their growth.

## Rer1 provides cytoprotection to support Myc-induced overgrowth

Our observation that the loser status of *rer1* mutant cells is due to the proteotoxic stress aligns with similar findings in *Rp* mutants [11–16,56]. However, counter-intuitively, the stress-induced UPR activation (PERK activity) in *rer1* mutants appears to be maladaptive (**Fig 4**), in contrast to the adaptive effect observed in Rp mutants [12]. We next sought to understand this discrepancy further. We reasoned that perhaps Rer1 itself is a component of the adaptive UPR response, providing cytoprotection. To test this, we turned to other UPR-dependent growth paradigms, such as Myc-driven overgrowth, which occurs under high proteotoxic stress and therefore it is highly dependent on the adaptive UPR pathways [33,35]. Thus, we explored the role of Rer1 in Myc-induced growth.

Using the GFP-Rer1 construct, we tested if Rer1 levels adapt to meet the increased proteostasis demand during Myc-overexpression. We found that overexpression of Myc in the posterior compartment via the *hh-GAL4* led to an increase in the GFP-Rer1 levels, indicating a potential connection between Rer1 and Myc-induced proteostasis demand (**Fig 7A–7B; quantified in Fig 7C**). Next, to evaluate the effect of Rer1 loss on proteostasis during Myc overexpression, we employed the ProteoStat aggresome detection assay, which identifies aggregation of misfolded protein. We found that RNAi-mediated depletion of Rer1 in the posterior compartment caused a modest increase in protein aggregation, which was also observed upon Myc overexpression (**Fig 7D–7F**). However, loss of Rer1 with Myc overexpression resulted in a substantial increase in the aggregation (**Fig 7G**; quantified in **Fig 7H**), suggesting that elevated Rer1 levels are required for proper proteostasis during Myc-overexpression.

We then asked if Rer1 is required for Myc-induced overgrowth. Thus, we induced MARCM clones overexpressing Myc in either control or *rer1$^{-/-}$*clones and analyzed the effect on clone size. Here, we observed that Myc-overexpression in otherwise normal cells increased growth of the clones, which was significantly restricted in the *rer1$^{-/-}$*cells (**Fig 8A–8D**; quantified in **Fig 8E**). Furthermore, p-eIF2α levels were strongly increased in Myc expressing *rer1$^{-/-}$*cell, as compared to either *rer1$^{-/-}$*or Myc-overexpression alone (**Fig 8A'–8D' and 8A'''–8D'''**; quantified in **Fig 8F**), indicating that the Rer1 alleviated proteotoxic stress in the Myc-overexpressing cell. However, these experiments did not rule out the possibility of an independent additive effect of Myc-overexpression and loss of Rer1 on p-eIF2α levels and clonal growth.

To further strengthen our results, we tested if reducing the overall dosage of Rer1 in the animal will affect Myc-overexpressing cells. To this end, we generated Actin-flip-out-GAL4 (AFG) Myc-overexpressing clones in either wild-type (*rer1$^{+/+}$*) or *rer1* heterozygous (*rer1$^{+/-}$*) background. Consistent with the observation in the MARCM clones, we find that

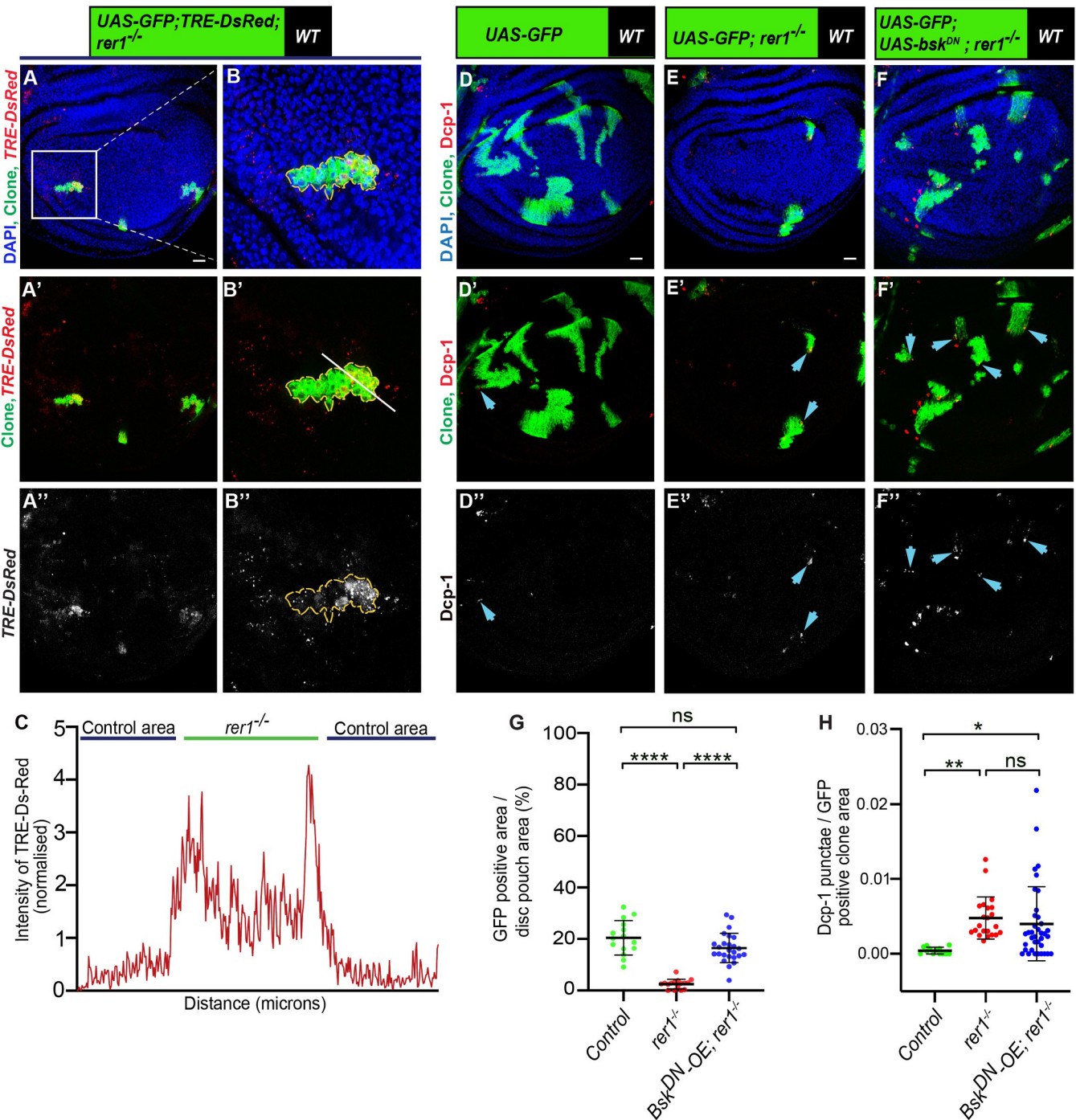

**Fig 6. JNK activity restricts the growth of *rer1*<sup>−/−</sup> clones independently of cell death.** (A-B) Representative images of hs-FLP-induced MARCM clones (72 hrs AHS) of *TRE-dsRED*, *UAS-GFP*, *rer1*<sup>−/−</sup> genotype (N = 17 wing discs). (C) Graph showing the intensity profile of *TRE-dsRED* along the line ROI (white) in **B'**. (D-F) Third-instar wing discs containing hs-FLP-induced MARCM clones (72 hrs AHS) of following genotypes, (D) *UAS-GFP (rer1*<sup>+/+</sup>*)*, (E) *UAS-GFP*, *rer1*<sup>−/−</sup>, and (F) *UAS-GFP*, *rer1*<sup>−/−</sup> + *UAS-bsk*<sup>DN</sup>, stained with the anti-Dcp-1 antibody. (G) Quantification of GFP-positive clone area in control discs (*UAS-GFP*; N = 15 wing discs), *rer1*<sup>−/−</sup>(*UAS-GFP*; N = 16 wing discs) and *bsk*<sup>DN</sup>-OE, *rer1*<sup>−/−</sup>(*UAS-GFP*, *UAS-bsk*<sup>DN</sup>; *rer1*<sup>−/−</sup>; N = 25 wing discs). Statistical analysis was performed using the Ordinary one-way ANOVA with Tukey's multiple comparison test (**** p<0.0001). (H) Quantification of the Dcp-1 in the GFP-labeled clones area in, **D** (N = 13 clones in 3 wing discs), **E** (N = 22 clones in 9 wing discs), and **F** (N = 37 clones in 12 discs). Statistical analysis was performed using the Ordinary one-way ANOVA with Tukey's multiple comparison test (** p = 0.0058,* p = 0.0154). SB = 20 μm. Also see **S10 Fig**.

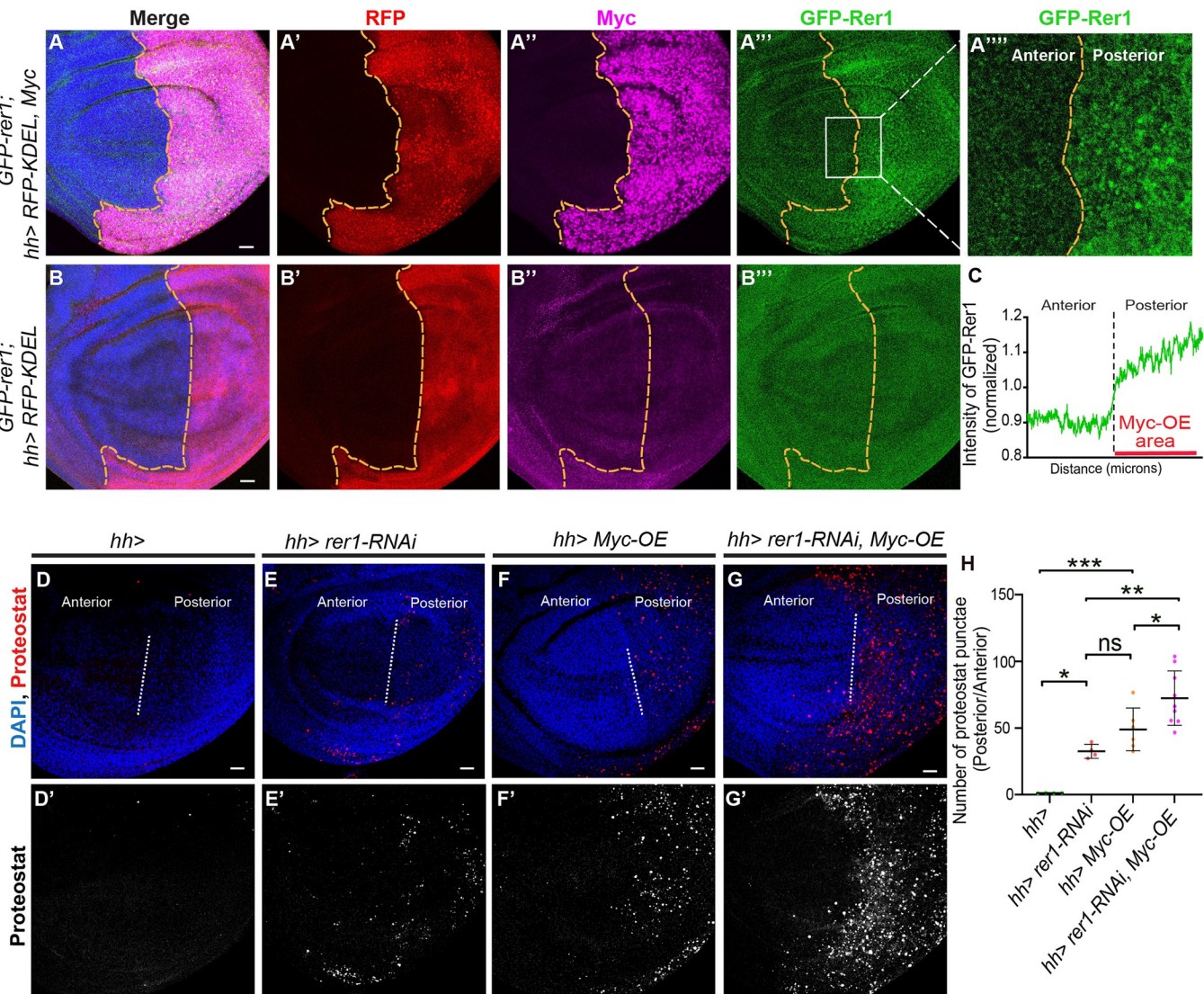

**Fig 7. Myc-overexpression increases Rer1 levels to maintain proteostasis. (A)** Images of wing discs with *hh-Gal4* driven Myc-overexpression, observed with anti-Myc staining (**A"**), in *GFP-rer1* background (*GFP-rer1; hh-Gal4::UAS-RFP-KDEL + UAS-Myc*; N = 10 wing discs) show higher levels of GFP-Rer1 in the posterior compartment as compared to the anterior. **(B)** The increase in GFP-Rer1 is not observed in the control discs (*GFP-rer1; hh-Gal4::UAS-RFP-KDEL*; N = 9 wing discs). **(C)** Quantification of GFP-Rer1 intensity in the area marked with white box in **A"'**, also enlarged in **A""**. **(D-G)** Protein aggregation levels were analysed with ProteoStat assay upon *hh-Gal4* mediated control (**D**; N = 4 wing discs), Rer1 knockdown (**E**; N = 4 wing discs), Myc overexpression (**F**; N = 6 wing discs) and overexpression of Myc along with Rer1 depletion (**G**; N = 9 wing discs). **(H)** Quantification of protein aggregation punctae in posterior compartment with respect to the anterior compartment in **D, E, F** and **G**. Statistical analysis was performed using the Ordinary one-way ANOVA with Tukey's multiple comparison test (*** p<0.0008,** p = 0.0024,* p = 0.0499 between **D** and **E**, * p = 0.0483 between **F** and **G**). SB = 20 μm. Anterior and posterior compartments of all wing imaginal discs were placed left and right sides, respectively.

overexpression of Myc in otherwise wild-type background led to an increase in p-eIF2α, which was further enhanced in the *rer1*$^{+/-}$ background (**Fig 9A–9B**; compare **Fig 9A" and 9B"**; quantified in **Fig 9C**). Moreover, the growth of Myc-expressing cells was reduced in the *rer1*$^{+/-}$ background as compared to the wild-type (**Fig 9A–9B**; quantified in **Fig 9D**). The control clones expressing only GFP did not affect the p-eIF2α levels and showed similar clonal growth in both wild-type and *rer1*$^{+/-}$ background (**S11A–S11D Fig**). Altogether, these results show that higher levels of Rer1 provided cytoprotection upon Myc-overexpression thereby supported the overgrowth.

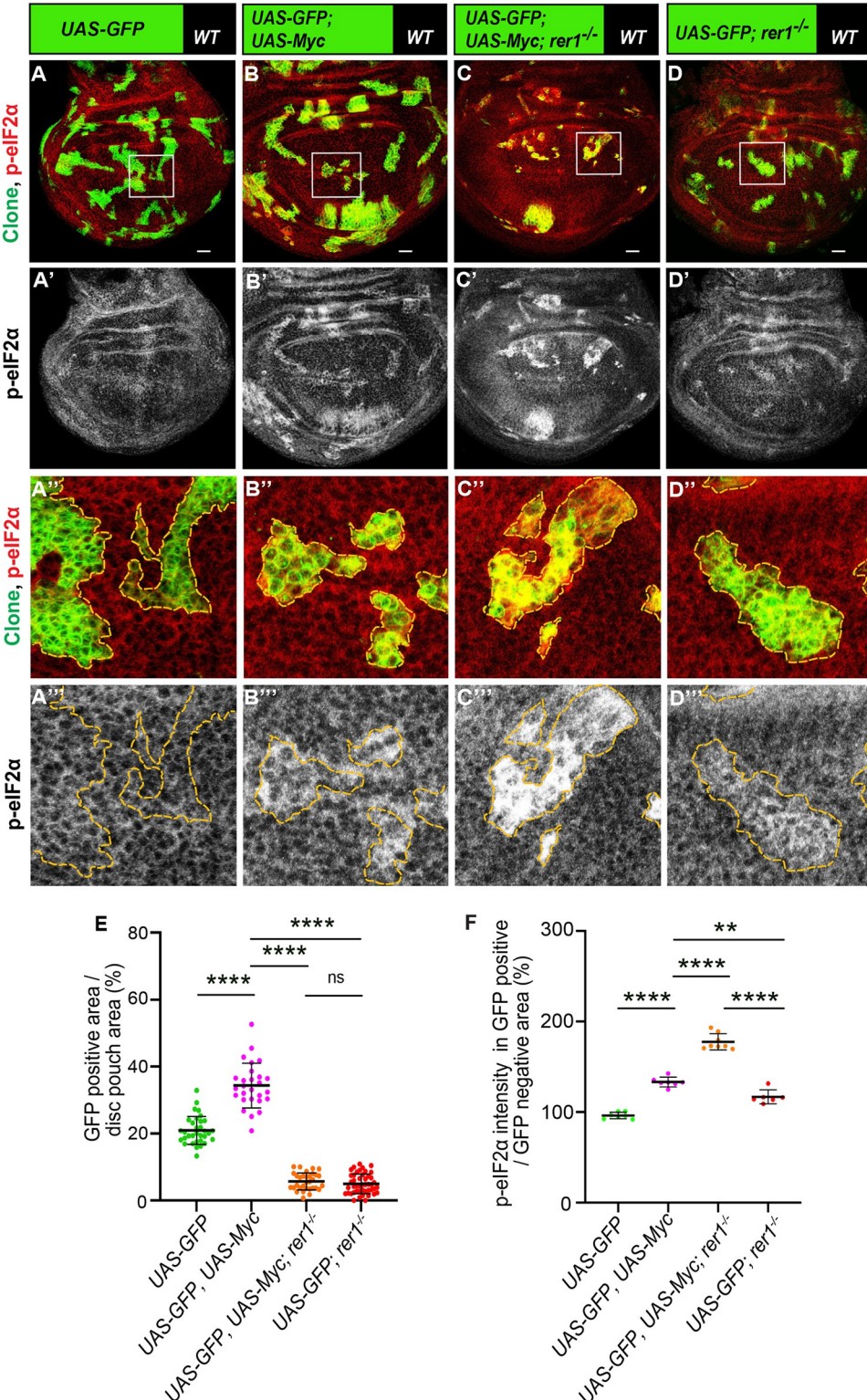

**Fig 8. Loss of Rer1 enhances proteotoxic stress in Myc-overexpressing cells. (A-D)** Representative images of the third-instar discs containing hs-FLP-induced MARCM clones (96 hrs AHS) of **(A)** *UAS-GFP (rer1$^{+/+}$)*, **(B)** *UAS-GFP; UAS-Myc*, **(C)** *UAS-GFP, rer1$^{-/-}$ + UAS-Myc* and, **(D)** *UAS-GFP, rer1$^{-/-}$* genotypes, stained with the anti-p-eIF2α antibody. **(A"-D")** Magnified image of the insets (white box) in **A, B, C** and **D** are shown in **A", B", C"** and **D"**, respectively. **(E)** Quantification of the relative size of GFP-labeled clones area in *UAS-GFP (rer1$^{+/+}$)* (N = 42 wing

discs), *UAS-GFP*, *UAS-Myc*, (N = 23 wing discs), *UAS-GFP*, *rer1*$^{-/-}$ + *UAS-Myc* (N = 31 wing discs) and, *UAS-GFP*, *rer1*$^{-/-}$ (N = 46 wing discs). **(F)** Quantification of the p-eIF2α levels inside the GFP-positive clones with respect to the nearby GFP-negative control tissue in genotypes given in, **A** (N = 7 wing discs), **B** (N = 7 wing discs), **C** (N = 8 wing discs) and **D** (N = 6 wing discs). Statistical analyses in **E** and **F** were performed using the Ordinary one-way ANOVA with Tukey's multiple comparison test (**** $p<0.0001$, ** $p = 0.0013$). SB = 20 μm.

## Discussion

In this study, we present compelling evidence supporting the role of Rer1 as a regulator of proteostasis in *Drosophila*. We show that Rer1 is localized to the ER and cis-Golgi compartments and loss of Rer1 activates PERK-mediated phosphorylation of eIF2α, indicating the onset of proteotoxic stress in the ER. These observations are consistent with the proposed function of Rer1 as a regulator of ER proteostasis [73]. Moreover, studies in yeast, worms and mouse cerebral cortex have shown that absence of Rer1 induces ER stress and activates the UPR pathways [43,74], suggesting a well-conserved function of Rer1 across species.

Our results show that proteotoxic stress induced by Rer1 deficiency creates loser cells that are eliminated via cell competition. This is in line with recent studies that have implicated mutations in genes, for example, RNA Helicase *Hel25E*, E3 ubiquitin ligase *Mahjong* and *Rp* mutations, in causing proteotoxic stress and subsequent elimination of mutant cells via cell competition [11,12,15,16,49,56].

Interestingly, our results show that despite both *rer1*$^{-/-}$ and *RpS3*$^{+/-}$ cells exhibiting increased p-eIF2α, *rer1* mutant cells outperform *RpS3*$^{+/-}$ cells when juxtaposed. Several potential reasons for the difference in their fitness can be envisaged. First, *RpS3*$^{+/-}$ cells, that are also *rer1*$^{+/-}$, may experience higher stress levels than *rer1*$^{-/-}$ cells, potentially rescuing later from their loser fate. Second, different downstream pathways may be activated by stress originating in the cytoplasm of *RpS3*$^{+/-}$ cells as compared to stress in the ER of Rer1-deficient cells. Third, the differential effect of p-eIF2α on translation was observed in *RpS3*$^{+/-}$ and *rer1*$^{-/-}$ cells. Our data shows that, despite having higher p-eIF2α, *rer1*$^{-/-}$ cells do not show reduction in translation. In contrast, *RpS3*$^{+/-}$ mutants show reduced translation attributed to the transcription factor Xrp1, that is both activated by stress and capable of promoting phosphorylation of eIF2α [15,16,56,75]. While the reduction in translation is probably not sufficient to induce cell competition [17], it may pose a disadvantage to the *RpS3*$^{+/-}$ cells when competing with the *rer1*$^{-/-}$ cells. In any case, aligned with other studies, our work shows that proteotoxic stress, even without a reduction in protein translation, is sufficient to drive elimination of the loser *rer1*$^{-/-}$ cells.

Stress-mediated activation of UPR is associated with both proapoptotic and cytoprotective responses [76,77]. However, whether or not the activation of UPR can provide cytoprotective support to the cell appears to depend on the initial cause of proteotoxic stress. For instance, in *RpS* mutants, p-eIF2α was shown to be cytoprotective [12], although not sufficiently to rescue their loser fate. In contrast, blocking p-eIF2α by PERK depletion was shown to suppress the competitive elimination of *wollknaeuel* (*wol*) mutant clones (wol is involved in the glycosylation of proteins in the ER), *Hel25E* mutant clones and *RpL14*$^{+/-}$ cells [15]. Consistent with this, we found that either PERK depletion or dephosphorylation of p-eIF2α by the expression of GADD34 improved the fitness of *rer1* mutant cells, indicating that these cells p-eIF2α played a maladaptive role. We propose that this is due to the involvement of Rer1 itself in the cytoprotective processes.

This was evident in our analysis of Rer1 in the regulation of proteostasis in Myc overexpressing cells and Myc-driven overgrowth. Myc-overexpression leads to PERK-mediated phosphorylation of eIF2α [30] and the induction of autophagy, which is required for the Myc-

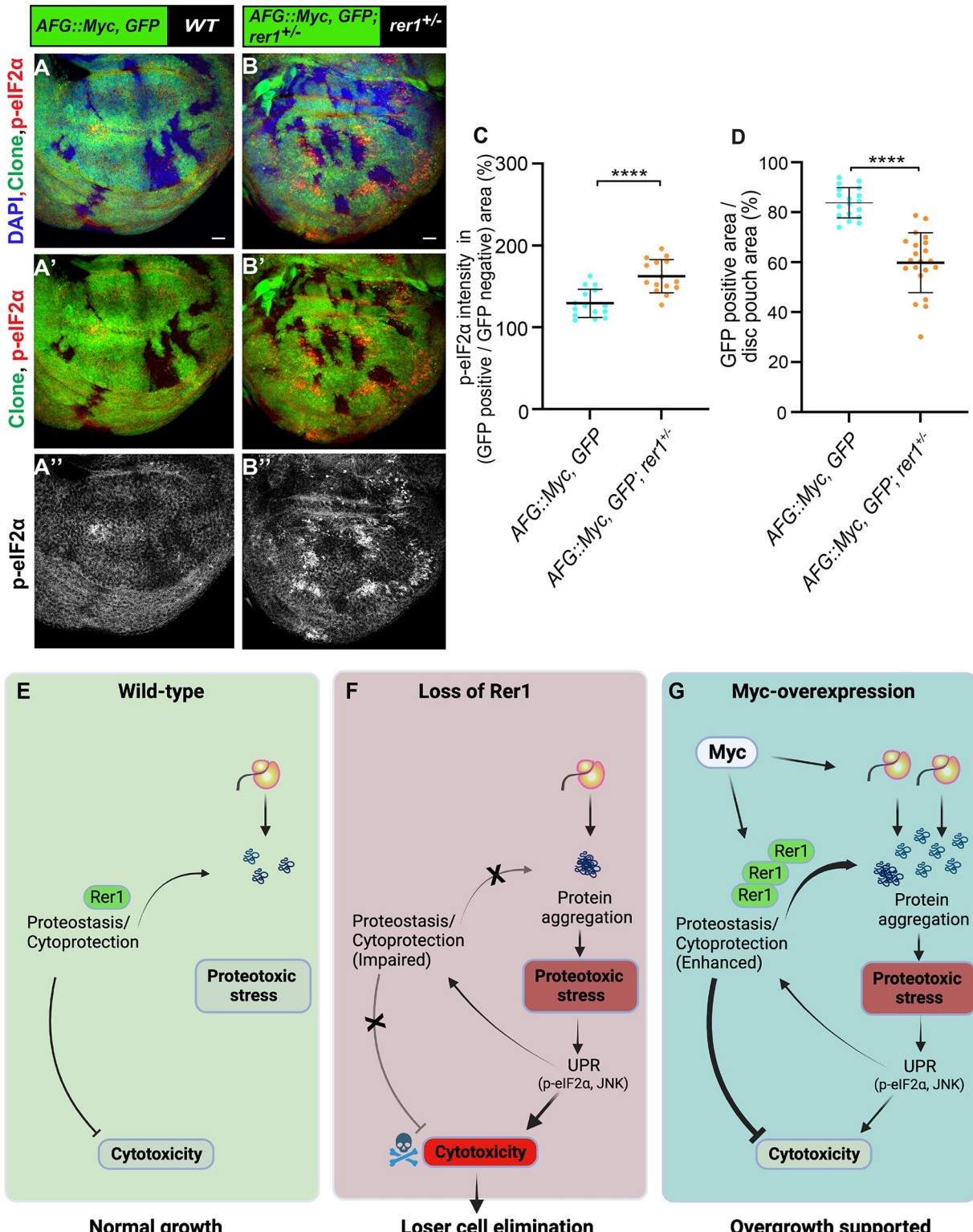

**Fig 9. Rer1 supports Myc-induced growth. (A-B)** Representative images of the wing imaginal discs with hs-FLP induced (at 48 hrs AEL) Actin-FRT-Stop-FRT-Gal4 (AFG)-clones overexpressing GFP and Myc (72 hrs AHS) in either wild-type **(A)** or *rer1*$^{+/-}$background **(B)**. **(C)** Quantification of p-eIF2α shows higher levels in *AFG:: Myc, GFP; rer1*$^{+/-}$(N = 15 wing discs) as compared to *AFG:: Myc, GFP* in WT background (N = 15 wing discs). **(D)** Quantification of GFP positive area shows reduction in *AFG:: Myc, GFP; rer1*+/–(N = 21 wing discs) as compared to *AFG:: Myc, GFP* in WT background (N = 18 wing discs). Statistical analyses in **C** and **D** were performed using the Two-tailed

Welch's unpaired t-test (\*\*\*\* p<0.0001). SB = 20 μm. **(E-G)** Schematic representation of the involvement of Rer1 in the regulation of proteostasis and its role in supporting Myc-induced overgrowth. **(E)** Rer1 plays a homeostatic role in the regulation of protein quality providing cytoprotection in the wild-type cells. **(F)** Cells lacking Rer1 have proteotoxic stress due to improper cytoprotection. This leads to high cytotoxicity downstream of UPR activation, ultimately leading to the competitive elimination of cells identified as losers. **(G)** Myc-overexpression increases gene expression leading to high proteotoxic stress. Rer1 levels are upregulated upon Myc-overexpression to maintain higher demand of proteostasis, thereby supporting overgrowth. Also see **S11 Fig**.

driven overgrowth [33]. This activation of UPR is believed to be an adaptive response to increased gene expression in Myc-overexpressing cells. The increase in p-eIF2α was shown to play a cytoprotective role which is also required for the growth of Myc-expressing tumors in mice [30,35]. We find that Rer1 levels are upregulated upon Myc-overexpression and loss of Rer1 further amplifies the proteotoxic stress and protein aggregation in Myc-overexpressing cells. Importantly, our data shows that the growth of Myc-overexpressing cells was suppressed in rer1$^{+/-}$background, where cells exhibited higher levels of p-eIF2α. These results suggest that higher levels of Rer1 allowed cells to maintain the increased demand for proteostasis upon Myc-overexpression and thereby mitigating the proteotoxic stress and supporting the overgrowth (**Fig 9E–9G**).

High Myc expression is associated with various cancers, for instance, pancreatic cancers [78]. However, targeting Myc protein directly as a therapeutic approach has not been successful so far [79]. Interestingly, besides Myc, Rer1 levels are also found to be high in pancreatic cancer cells [80]. Thus, further studies on the effect of Rer1 in Myc-induced overgrowth will be helpful in developing Rer1 as a potential therapeutic target in Myc-driven cancers.

## Materials and methods

### *Drosophila* genetics and culturing

The following *Drosophila* stocks were used: *neoFRT82B ry506 rer1$^{KO}$* on 3rd chr. (this paper); *GFP-Rer1* genomic rescue line on 2nd chr (this paper), *hh-Gal4* on 3rd chr. [81], *hs-FLP* on 1st chr. and *UAS-GFP* on 3rd chr. were gifted by A. Teleman (DKFZ, Germany). The following additional stocks were obtained from the Bloomington *Drosophila* stock center (BDSC)*; neoFRT82B Ubi-mRFP.nls (BDSC# 30555), neoFRT82B Ubi-GFP.nls (BDSC# 32655), FRT82B Ubi-GFP.nls, RpS3[Plac92] (BDSC# 5627), hs-FLP UAS-GFP tubP-Gal4;; neoFRT82B tubP-Gal80 (BDSC# 86311), UAS-Trip-rer1-RNAi (BDSC# 57435), UAS-GADD34 (BDSC# 76250), UAS-dMyc (BDSC# 9674), hid-LacZ (BDSC# 57435), UAS-bsk$^{DN}$ (BDSC# 6409), UAS-RFP. KDEL (BDSC# 30909), TRE-DsRed (BDSC# 59011), UAS-GCN2 RNAi (BDSC# 67215), UAS-PERK RNAi (BDSC# 42499)* and *Actin5C-FRT-CD2-FRT-Gal4 (BDSC# 4779)*. The following line was obtained from the Vienna *Drosophila* RNAi center; *UAS-rer1-RNAi (GD ID# 23204)*.

Standard food composition containing cornmeal, sucrose, yeast, Dextrose and agar was used for growing all fly cultures and crosses. All crosses were maintained at 25˚C room temperature unless specifically mentioned. Egg collection, heat shock, dissection (time points), immunostaining, and imaging were kept identical between the control and experiments.

### Generation of *rer1* mutant and genomic rescue fly-lines

To generate the *rer1* knock-out line, the fly line containing a P-element insertion upstream of the *rer1* coding sequence (*y; ry506 P{SUPor-P}CG11857KG08816/TM3, Sb Ser*) was obtained from the Bloomington *Drosophila* Stock Center Indiana (BDSC# 15137). The P-element imprecise excision was performed, and the excision lines were screened by Polymerase chain reaction (PCR), using primers positioned in the terminal coding sequences of the neighboring genes. The *rer1* knock-out line, *w;; neoFRTneo82B, ry506, rer1$^{KO}$/TM6C, Sb, Tb* was identified,

with 1560bp deletion that covered the whole coding sequence of the *rer1* gene (Fig 1A). To generate the *GFP-rer1 (CG11857) BAC* rescue construct, GFP was recombined to *CG11857BAC* genomic clone, *CH322-101C14* (BACPAC resources) following the P[acman] method [82]. The *GFP-CG11857BAC* was then inserted into *VK18* (2L, 53B2) for genomic rescue.

## Antibodies

Larval wing imaginal discs were stained using the following antibodies: Rabbit anti-cleaved Dcp-1 (1:300, Cell Signaling Technology), Rabbit-anti-p-eIF2α (1:300, Cell Signaling Technology), Mouse-anti-dMyc (1:20, a gift from Bruce Edgar lab, University of Utah, USA), Mouse-anti-beta Galactosidase (1:50, 40-1A, DSHB) and Hoechst 33342, H3570 (1:1000, Invitrogen). Fluorescent secondary antibodies used were Alexa-405, Alexa-488, Alexa-594, and Alexa-647 (Invitrogen) at 1:500 dilutions.

## ProteoStat assay

For ProteoStat staining, larvae were dissected in 1X PBS and transferred to an Eppendorf tube containing 4% formaldehyde diluted in 1X PAB (ProteoStat assay buffer) for 30 minutes. Following that, the samples were permeabilized 3 times using 0.5% Triton X-100 and 3 mM ethylenediaminetetraacetic acid (pH 8.0) diluted in 1X PAB. Next, it was stained with a ProteoStat detection reagent (Enzo Life Sciences) diluted 1 in 20,000 and Hoechst 33342 at 1 μg ml$^{-1}$ in PAB incubated for 45 minutes at 4˚C and washed thrice by PBS. The samples were then mounted and imaged immediately using a confocal microscope.

## Immunostaining

Third instar wandering larvae were used for immunohistochemistry. Larvae were dissected in 1X Phosphate-buffered saline (PBS) and head complexes with wing imaginal discs were fixed for 45 min in 4% paraformaldehyde (PFA) at room temperature. Wing discs were blocked with 0.1% BSA for 1 hour followed by overnight incubation with primary antibody at 4˚C and 90 min incubation with fluorophore-conjugated secondary antibody at room temperature. Staining and microscopy conditions for samples used were identical. Wing discs are oriented with the dorsal up and anterior left.

## Image acquisition and processing

Images of fixed samples were acquired using the 40x oil objective on the Olympus (FV3000) confocal microscope with each slice (z-stack) equivalent to 1μm. Wing disc images were processed using ImageJ (ImageJ version 1.51j8, NIH) and Photoshop (Adobe Photoshop CS6 extended version 13.0 x64). Figures were made using Illustrator (Adobe Illustrator CS6 Tryout version 16.0.0). All the schematics were created online with *BioRender.com*.

## Statistical analysis

Wing disc pouch regions were taken for all the analysis. Clone size was measured as "total clone area per disc pouch area (%)" by analyzing all of the clones in the pouch area of each genotype using ImageJ (ImageJ version 1.51j8, NIH) software. For cell death analysis, Dcp1 punctae were counted in the tissues (as mentioned in the Figure captions) and normalized with their respective clone area. To analyze the position of cell death in clones, Dcp1 punctae at the clone border and center were counted and divided with the respective common clone area. To calculate the fold change, the average pixel intensities within the area of interest were

divided by that of the corresponding wild-type area. All raw data were analyzed using Excel (Microsoft) and graphs were plotted using GraphPad (GraphPad Prism 8 version 8.0.2). Each raw data set is shown as a dot plot, and the horizontal line represents the median.

Statistical analyses were performed by using unpaired two-tailed Welch's t-test to compare areas in Figs 2E, 9C–9D, S3D and S11D, cell death in S1I–S1J Fig, adult wing area in S1O and S2E Figs, OPP intensity in S9C Fig, pupariation time in S10G Fig and p-eIF2α intensity in S11C Fig. Paired two-tailed Wilcoxon signed-rank test was used to compare the cell death between center and border of each clone area in Figs 1K, 2F, and S3C. For multiple comparisons, Ordinary one-way analysis of variance (ANOVA) with Tukey's test was performed to analyze the data set in Figs 1E, 1J, 4E–4F, 4K, 5C, 5F–5G, 6G–6H, 7H, 8E–8F or ANOVA-Dunnett's test to compare the survivals of $rer1^{-/-}$ and rescued $rer1^{-/-}$ flies with control ($rer1^{+/+}$) flies in S1B Fig. The significance level was set to $p < 0.05$. No statistical methods were used to predetermine the sample size. For the quantification of survival and development, first instar larvae were collected, and the numbers of survival larvae in different developmental stages were counted each day. All experiments were independently performed at least three times and were not randomized or blinded. Source data for all the quantitative analyses is included in S1 Data.

## Generation of clones

For genetic mosaic analysis, the FLP (Flippase)/FRT (Flippase recognition target) system [83] was used to generate mosaic clones in the wing imaginal disc. Heat shock-inducible FLP was expressed for the mitotic recombination in both mitotic clones and MARCM clones [84,85]. Heat shock was given in a 37˚C water bath for 60 minutes at 48 hrs AEL (after egg laying) and larvae were then shifted to 25˚C. Larvae were dissected at 72 hrs or 96 hrs or 120 hrs AHS (After heat-shock), as mentioned specifically. Similar heat shock strategy was used for the AFG clones shown in Figs 9 and S11, and larvae were dissected 72 hrs AHS kept at 25˚C.

## Drosophila genotypes

The following genotypes were used in this study:

Fig 1B–1D: *hs-FLP/+;; FRT82B, Ubi-RFP.nls/ neoFRT82B, ry^{506}, rer1^{KO}*

Fig 1F–1F': *hs-FLP/+;; FRT82B, Ubi-RFP.nls/ FRT82B, Ubi-GFP.nls*

Fig 1G–1G', 1H–1H': *hs-FLP/+;; FRT82B, Ubi-RFP.nls/ neoFRT82B, ry^{506}, rer1^{KO}*

Fig 1I–1I": *hs-FLP/+; GFP-rer1/+; FRT82B, Ubi-RFP.nls/ neoFRT82B, ry^{506}, rer1^{KO}*

Fig 2A–2A", 2B–2B": *hs-FLP/+;; FRT82B, Ubi-RFP.nls/ FRT82B, Ubi-GFP.nls, RpS3[Plac92]*

Fig 2C–2C", 2D–2D": *hs-FLP/+;; FRT82B, Ubi-GFP.nls, RpS3[Plac92]/ neoFRT82B, ry^{506}, rer1^{KO}*

Fig 3A–3A''', 3B–3B'': *hs-FLP/+;; FRT82B, Ubi-RFP.nls/ neoFRT82B, ry^{506}, rer1^{KO}*

Fig 3D–3D''', 3E–3E''': *hs-FLP/+; GFP-rer1/+; FRT82B, Ubi-RFP.nls/ neoFRT82B, ry^{506}, rer1^{KO}*

Fig 4A–4A''', 4G–4G''': *hs-FLP, UAS-GFP/+;; tubP-Gal4, neoFRT82B, tubP-Gal80/ neoFRT82B, Ubi-mRFP.nls*

Fig 4B–4B''', 4H–4H''': *hs-FLP, UAS-GFP/+;; tubP-Gal4, neoFRT82B, tubP-Gal80/ neoFRT82B, ry^{506}, rer1^{KO}*

Fig 4C–4C''', 4I–4I''': *hs-FLP, UAS-GFP/+; UAS PERK-RNAi/+; tubP-Gal4, neoFRT82B, tubP-Gal80/ neoFRT82B, ry^{506}, rer1^{KO}*

Fig 4D–4D''', 4J–4J''': *hs-FLP, UAS-GFP/+; UAS GCN2 RNAi /+; tubP-Gal4, neoFRT82B, tubP-Gal80/ neoFRT82B, ry^{506}, rer1^{KO}*

Fig 5A–5A", 5B–5B", 5D–5D", 5E–5E": *hs-FLP, UAS-GFP/+; UAS-GADD34/+; tubP-Gal4, neoFRT82B, tubP-Gal80/ neoFRT82B, ry$^{506}$, rer1$^{KO}$*

Fig 6A–6A", 6B–6B": *hs-FLP, UAS-GFP/+; TRE-DsRed/+; tubP-Gal4, neoFRT82B, tubP-Gal80/ neoFRT82B, ry$^{506}$, rer1$^{KO}$*

Fig 6D–6D": *hs-FLP, UAS-GFP/+;; tubP-Gal4, neoFRT82B, tubP-Gal80/ neoFRT82B, Ubi-mRFP.nls*

Fig 6E–6E": *hs-FLP, UAS-GFP/+;; tubP-Gal4, neoFRT82B, tubP-Gal80/neoFRT82B, ry$^{506}$, rer1$^{KO}$*

Fig 6F–6F": *hs-FLP, UAS-GFP/UAS-bsk$^{DN}$; +/+; tubP-Gal4, neoFRT82B, tubP-Gal80/ neoFRT82B, ry$^{506}$, rer1$^{KO}$*

Fig 7A–7A"": *;; GFP-rer1/UAS-Myc; hh-Gal4, UAS-RFP-KDEL/+*

Fig 7B–7B"': *;; GFP-rer1/+; hh-Gal4, UAS-RFP-KDEL/+*

Fig 7D–7D': *;;; hh-Gal4/+*

Fig 7E–7E': *;;; UAS-rer1-RNAi, hh-Gal4/+*

Fig 7F–7F': *;; UAS Myc/+; hh-Gal4*

Fig 7G–7G': *;; UAS Myc/+; UAS-rer1-RNAi, hh-Gal4/+*

Fig 8A–8A"': *hs-FLP, UAS-GFP/+;; tubP-Gal4, neoFRT82B, tubP-Gal80/ neoFRT82B, Ubi-mRFP.nls*

Fig 8B–8B"': *hs-FLP, UAS-GFP/+; UAS-Myc/+; tubP-Gal4, neoFRT82B, tubP-Gal80/ neoFRT82B, Ubi-mRFP.nls*

Fig 8C–8C"': *hs-FLP, UAS-GFP/+; UAS-Myc/+; tubP-Gal4, neoFRT82B, tubP-Gal80/ neoFRT82B, ry$^{506}$, rer1$^{KO}$*

Fig 8D–8D"': *hs-FLP, UAS-GFP/+;; tubP-Gal4, neoFRT82B, tubP-Gal80/ neoFRT82B, ry$^{506}$, rer1$^{KO}$*

Fig 9A–9A": AFG/*hs-FLP; UAS-Myc/+; UAS-GFP/+*

Fig 9B–9B": AFG/*hs-FLP; UAS-Myc/+; UAS-GFP/neoFRT82B, ry$^{506}$, rer1$^{KO}$*

## Supporting information

**S1 Fig. (Supporting main Fig 1): Analysis of *rer1$^{KO}$ (rer1$^{−/−}$)* and *rer1*-RNAi in growth and cell survival. (A)** *rer1* mRNA expression levels were measured by quantitative PCR in *rer1$^{KG08816}$* (control, precise excision) and homozygous *rer1$^{KO}$ (rer1$^{−/−}$)* flies. Bars show mean ±SEM (N = 3 independent experiments). **(B)** *rer1$^{−/−}$* flies failed to hatch out. Most of them died before the pupae stage. Re-introducing the *rer1* genomic fragment (*GFP-rer1*) rescued the phenotype, underscoring that they are caused by *rer1* deficiency. Statistical analyses in **B** were performed using the Ordinary one-way ANOVA with Dunnett's multiple comparison test, (**** $p < 0.0001$). **(C)** Scheme of a wing disc illustrating anterior and posterior compartments in which transgene was expressed with a posterior specific Gal4. **(D)** *hh-Gal4* mediated depletion of Rer1 in *GFP-rer1* background [*GFP-rer1; hh-Gal4::UAS-rer1-RNAi*] shows loss of GFP signal in the posterior compartment. **(E–F)** Dcp-1 staining on **(E-E')** control (*hh-Gal4*, N = 7 wing discs) and **(F-F')** Rer1 depleted (*hh-Gal4, rer1*-RNAi, N = 21 wing discs) wing discs. **(G–H)** Acridine Orange (AO) staining on **(G-G')** control (*hh-Gal4*, N = 5 wing discs) and **(H-H')** Rer1 depleted (*hh-Gal4, rer1*-RNAi, N = 14 wing discs) wing discs. **(I-J)** Quantification of Dcp-1 punctae **(I)** and AO punctae **(J)** numbers in the posterior compartments of either control or Rer1 depleted discs, normalized to their respective anterior compartments. Statistical analysis was performed using two-tailed Welch's t-test (**** $p < 0.0001$). SB = 20 μm. **(K-N)** Adult wings from control flies (*hh-Gal4::2x UAS-GFP*); male, **K**, N = 12 and female, **M**, N = 13) and flies harboring *hh-Gal4* mediated Rer1 knockdown along with overexpression of GFP (*hh-Gal4::UAS-rer1-RNAi, UAS-GFP);* male, **L**, N = 11 and female, **N**, N = 22). **(O)**

Quantification of the adult wing areas measured within the dotted line. Statistical analysis was performed using two-tailed Welch's t-test. P values for male wing comparison **(K-L)** was *p = 0.0793*, for female wing comparison **(M-N)** was *p = 0.6704*. SB = 500 μm.
(TIF)

**S2 Fig. (Supporting main [Fig 1]): Wing size is unaffected upon induction of *rer1* mutant clones albeit higher cell death. (A-B)** Images of adult wings from control flies harboring wild-type clones induced at 48hrs AEL (**A,** male of genotype *hs-FLP/Y;; FRT82B, Ubi-RFP.nls/ FRT82B, Ubi-GFP.nls*; N = 39 and **B,** female of genotype *hs-FLP/+;; FRT82B, Ubi-RFP.nls/ neoFRT 82B, ry$^{506}$, rer1$^{KO}$*; N = 44). **(C-D)** Images of adult wings from flies with *rer1* mutant clones induced at 48hrs AEL (**C**, males of genotype *hs-FLP/ Y;; FRT82B, Ubi-RFP.nls/ FRT82B, Ubi-GFP.nls*; N = 20 and **D**, females of genotype *hs-FLP/+;; FRT82B, Ubi-RFP.nls/ neoFRT82B, ry$^{506}$, rer1$^{KO}$*; N = 25). **(E)** Quantification of the adult wing area measured within the dotted line (Two-tailed Welch's t-test). P values for male wings size comparison **(A-C)** was *P = 0.1494*, for female wings size comparison **(B-D)** was *P = 0.5407*. SB = 500 μm. **(F-G)** Wing imaginal disc harboring RFP negative *rer1$^{-/-}$*clones (72 hrs AHS), stained with anti Dcp-1 antibody to show the cell death at clone boundary. **(G)** A magnified image of the white inset in **F**. SB = 20 μm.
(TIF)

**S3 Fig. (Supporting main [Fig 2]): MARCM clones expressing *rer1*-RNAi show cell death at the boundary. (A-A")** Representative images of the wing discs containing hs-FLP induced *rer1-RNAi* expressing MARCM clones (72 hrs AHS), stained with anti-Dcp-1 antibody. **(B-B")** Magnified images of the white box in **A**. **(C)** Quantification of Dcp-1 positive cells at the center and border of *rer1-RNAi* clones (N = 29 clones in 13 wing discs); two-sided Wilcoxon signed-rank test, **** p<0.0001. **(D)** Quantification of GFP positive clone area in wing disc harboring wild-type (N = 11 wing discs) and *rer1*-RNAi (N = 16 wing discs) MARCM clones (Two-tailed Welch's t-test). SB = 20 μm. *** p = 0.0009.
(TIF)

**S4 Fig. (Supporting main [Fig 3]): Colocalization of GFP-Rer1 with ER and Golgi. (A-A''')** Colocalization of GFP-tagged Rer1 and ER, stained with Calnexin (red). **(B-B''')** Colocalization of GFP-tagged Rer1 and Golgi, marked by the Golgin-245 (red). White arrows showed the colocalized punctae. SB = 20 μm.
(TIF)

**S5 Fig. (Supporting main [Fig 3]): Knockdown of Rer1 leads to proteotoxic stress. (A)** Control wing disc (*hh-Gal4::UAS-GFP, N = 5)* showing expression of *hh-Gal4* in the posterior compartment, marked by GFP. **(B–D)** *hh-Gal4* mediated depletion of Rer1 (*hh-Gal4::UAS-GFP, UAS-rer1-RNAi*) in the posterior compartment using three different RNAi lines *23203/GD* (**B**; *N = 6 wing discs); 23204/GD* (**C**; *N = 8 wing discs) and 57435/Trip* (**D**; *N = 6 wing discs*), shows an increase in the p-eIF2α level compare to the anterior compartment. **(E-F)** Images of the wing discs containing hs-FLP induced *rer1*-RNAi (Trip) expressing MARCM clones (72 hrs AHS; N = 3 wing discs), stained with anti-p-eIF2α antibody. **(F-F")** Magnified images of the boxed area in **E**. SB = 20 μm.
(TIF)

**S6 Fig. (Supporting main [Fig 3]): Knockdown of Rer1 results in the accumulation of ROS. (A-D)** DHE uptake assay, indicative of ROS levels, performed on the control and Rer1 depleted wing imaginal discs. **(A)** Depletion of Rer1 in the posterior compartment *[hh-Gal4:: UAS-rer1-RNAi]* showed higher levels of DHE as compared to the anterior compartment

(N = 12 wing discs). **(B)** Magnified images of the inset in **A**. **(C)** control disc [*hh-Gal4*] showed similar DHE levels between anterior and posterior compartments (N = 8 wing discs). **(D)** Magnified images of the inset in **C**. Yellow dotted lines mark the anterior-posterior boundary; Nuclei are stained with DAPI. SB = 20 μm.
(TIF)

**S7 Fig. (Supporting main [Fig 4]): Loss of Rer1 triggers PERK-mediated phosphorylation of eIF2α. (A-A')** Control third-instar wing disc (*hh-Gal4::2X UAS-GFP*) stained with anti-p-eIF2α antibody (N = 10 wing discs). **(B-D)** Third-instar discs with *hh*-Gal4 mediated coexpression of GFP with *PERK*-RNAi (**B-B'**; N = 12 wing discs), *GCN2*-RNAi (**C-C'**; N = 5) and *rer1*-RNAi (**D-D'**; N = 10 wing discs), stained with anti-p-eIF2α antibody. **(E-E')** Disc with coexpression of *rer1*-RNAi and *PERK*-RNAi, stained with anti-p-eIF2α antibody (N = 12). **(F-F')** Disc with coexpression of *rer1*-RNAi and *GCN2*-RNAi, stained with anti-p-eIF2α antibody (N = 13). Nuclei are stained with DAPI. SB = 20 μm.
(TIF)

**S8 Fig. (Supporting main [Fig 5]): Overexpression of GADD34 rescued cell death following Rer1 depletion. (A-B)** Third instar disc with *hh-Gal4* mediated overexpression of GADD34 and expression of *rer1*-RNAi, stained with either anti-p-eIF2α (**A**; N = 10 wing discs) and anti-cleaved Dcp-1 antibodies (**B**; N = 15 wing discs). SB = 20 μm.
(TIF)

**S9 Fig. Protein translation is unaffected upon loss of Rer1. (A-B)** OPP assay on third-instar discs containing hs-FLP-induced MARCM clones (96 hrs AHS) of **(A)** *UAS-GFP (control)* and **(B)** *UAS-GFP, rer1⁻/⁻* genotypes. **(C)** Quantification of the signal intensity of OPP inside the GFP-positive clones with respect to nearby GFP-negative control tissue in *UAS-GFP* (**A**; N = 3 wing discs), *UAS-GFP, rer1⁻/⁻* (**B**; N = 5 wing discs). Borders between the GFP-positive and GFP-negative areas are marked with yellow dotted lines. Statistical analysis in **C** was performed using the two-tailed Welch's t-test (p = 0.9431). SB = 20 μm.
(TIF)

**S10 Fig. (Supporting main [Fig 6]): Loss of Rer1 activates JNK signaling. (A-B)** Images representing *puc-lacZ* promoter activities via beta-galactosidase (red) staining on **(A-A')** control discs (*puc-lacZ; hh-Gal4/+;* N = 9 wing discs*)*, or **(B-B')** Rer1 depleted discs *puc-lacZ; hh-Gal4, UAS-rer1-RNAi;* N = 10 wing discs). **(C)** Third-instar wing discs containing hs-FLP-induced (96 hrs AHS) MARCM clones of *UAS-GFP, rer1⁻/⁻* genotype (N = 4 wing discs), immunostained for the beta-galactosidase (red) to mark the *hid-lacZ* promoter activity. **(D-D')** A magnified image of the inset (white box) in **C**. The interfaces between the clone areas are marked with yellow dotted lines. **(E-F)** Representative images of hs-FLP-induced MARCM clones (72 hrs AHS) of *UAS-GFP, UAS-bsk^{DN}; rer1⁻/⁻* genotype stained with anti-p-eIF2α antibody (N = 6 wing discs)**. (G)** Quantification of pupariation time for *UAS-GFP; rer1⁻/⁻* (N = 16 technical repeats) and *UAS-GFP, UAS-bsk^{DN}; rer1⁻/⁻* (N = 16 technical repeats). Statistical analysis in **G** was performed using the two-tailed Welch's t-test (p = 0.1389). SB = 20 μm.
(TIF)

**S11 Fig. (Supporting main [Fig 9]): Analysis of GFP expressing control AFG clones.** (A-B) Representative images of the wing imaginal discs with hs-FLP induced (48 hrs AEL) Actin-FRT-Stop-FRT-Gal4 (AFG)-control clones overexpressing GFP in either wild-type (A) or *rer1⁺/⁻* background (B), dissected 72 hrs AHS. (C-D) Quantification of p-eIF2α and GFP positive area in AFG:: GFP in WT background (N = 11 wing discs) and AFG:: GFP in *rer1⁺/⁻* background (N = 5 wing discs). Statistical analyses in C and D were performed using the Two-

tailed Welch's unpaired t-test (p = 0.4193 and p = 0.6089, respectively). SB = 20 μm.
(TIF)

**S1 Text. Supplemental Experimental Procedure.**
(DOCX)

**S1 Data. Quantitative data used for graphs presented in Figs 1–9, S1–S3Figs and S9–S11 Figs.**
(XLSX)

## Acknowledgments

We thank IISER Bhopal for the fly facility and the DST-FIST facility for the confocal microscopy. The authors thank Milos Spasic (now at GSK, Belgium) for generating the *rer1* mutant flies. We thank Dr. Vimlesh Kumar (IISER Bhopal) for comments on the manuscript. We thank V.C. group members for the discussions on the project and manuscript.

## Author Contributions

**Conceptualization:** Pranab Kumar Paul, Varun Chaudhary.

**Data curation:** Pranab Kumar Paul, Shruti Umarvaish.

**Formal analysis:** Pranab Kumar Paul, Shruti Umarvaish, Shivani Bajaj, Rishana Farin S., Hrudya Mohan, Wim Annaert.

**Funding acquisition:** Wim Annaert, Varun Chaudhary.

**Investigation:** Pranab Kumar Paul, Shruti Umarvaish, Shivani Bajaj, Rishana Farin S., Hrudya Mohan, Wim Annaert.

**Methodology:** Pranab Kumar Paul, Shruti Umarvaish, Shivani Bajaj, Rishana Farin S., Wim Annaert.

**Project administration:** Varun Chaudhary.

**Resources:** Wim Annaert, Varun Chaudhary.

**Supervision:** Varun Chaudhary.

**Validation:** Pranab Kumar Paul, Shruti Umarvaish, Shivani Bajaj, Hrudya Mohan.

**Visualization:** Pranab Kumar Paul, Shruti Umarvaish, Shivani Bajaj, Rishana Farin S., Hrudya Mohan.

**Writing – original draft:** Pranab Kumar Paul, Varun Chaudhary.

**Writing – review & editing:** Pranab Kumar Paul, Shruti Umarvaish, Shivani Bajaj, Wim Annaert, Varun Chaudhary.

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
