## [Decision Letter · Decision Letter 0]

3 Jul 2023

Dear Dr Chaudhary,

Thank you very much for submitting your Research Article entitled 'Drosophila Rer1 is essential for the maintenance of protein homeostasis and Myc-driven super-competition' to PLOS Genetics.

The manuscript was fully evaluated at the editorial level and by independent peer reviewers. The reviewers appreciated the attention to an important problem, but raised some substantial concerns about the current manuscript. Based on the reviews, we will not be able to accept this version of the manuscript, but we would be willing to review a much-revised version.  The reviewers suggest additional experiments to strengthen the manuscript. I agree with their assessment. While I realize these experiments may take a significant amount of time to perform, the additional information they would provide will be essential for a more positive review.  We cannot, of course, promise publication at that time.

If you decide to revise the manuscript for further consideration at PLOS Genetics, please aim to resubmit within the next 60 days, unless it will take extra time to address the concerns of the reviewers, in which case we would appreciate an expected resubmission date by email to plosgenetics@plos.org.

We are sorry that we cannot be more positive about your manuscript at this stage. Please do not hesitate to contact us if you have any concerns or questions.

Yours sincerely,

Ken M. Cadigan

Academic Editor

PLOS Genetics

Gregory P. Copenhaver

Editor-in-Chief

PLOS Genetics

Reviewer's Responses to Questions

**Comments to the Authors:**

Reviewer #1: Paul et al present a manuscript on the role of Rer1 role in protein homeostasis and Myc-driven super-competition. Although the manuscript compiles interesting data, the study has limitations that make it difficult to recommend for publication. There are three important results. First, Rer1 is an essential gene in Drosophila. Second, Rer1 mutant cells are eliminated from wing epithelium via cell competition. Lastly, Rer1 is required for Myc-driven cell competition. Currently, an in-depth study of any of the three points is missing. Addressing the below-mentioned points would help authors to improve their manuscript:

1. As per the title of the manuscript, the major claim of the authors is that Rer1 is required for the maintenance of protein homeostasis. The supporting evidence is that Rer1 localizes in Golgi & ER and loss of Rer1 results in increased p-eIf2 α levels. A more detailed study is required here to dissect if Rer1 has a direct role in the regulation of protein homeostasis or if p-eif2α levels increase because of activation of proteotoxic stress signaling in Rer1 mutant, as shown recently in ribosome heterozygous cells or mahjong mutant cells (Langton et al 2021 PLoS genetics). Moreover, does the Rer1 mutant accumulate unfolded proteins? Also, does PERK or GCN2 kinase activity go up or does the dephosphorylation of p-eIf2alpha go down? Perturbation of protein homeostasis also affects autophagic and proteasomal flux, addressing all these points will make the study more comprehensive.

2. Regarding the role of Rer1 in cell competition, the data presented in Fig 2 A-E does not support that Rer1 has a role in cell competition. This data could also be possible because of protein perdurance in somatic clones and mutant cells display cell autonomous cell death.

3. The current model for competitive elimination of RpS3 heterozygous is through Xrp1. Xrp1 regulates proteotoxic stress signaling in RpS3 cells (Langton et al 2021, PLoS Genetics). Thus, data related to the continuous elimination of RpS3/+ cells in the presence of Rer1 mutant, as shown in Fig 2F-I, shows that most likely Xrp1 is still activated in Rp3/+ cells. Interestingly, Xrp1 is known to be active in RpS3/+ cells in a cell-autonomous manner (Lee et al 2018, Dev cell). Therefore, how this data supports the role of Rer1 in cell competition is not clear.

4. The data presented in Fig 1B-E does support the role of Rer1 in cell competition. However, this data need to be validated by knocking down Rer1 through its RNAi and then showing boundary cell death and clone area compared to control clones.

5. The model suggested for Rer1 cell competition would be more complete by studying genetic epistasis between p-eIf2α and JNK signaling in Rer1 mutant cells.

6. p-eIf2α role in cell competition is already known (Naotaka Ochi et al 2021 PLoS Genetics) and it plays a cytoprotective role in different stress conditions. Therefore, how its upregulation results in the elimination of Rer1 is not clear. Moreover, it would be interesting to examine if GADD34 overexpression rescues cell autonomous cell death that occurs upon knockdown of Rer1 in the posterior compartment. This will help to dissect if p-eIf2α has a cell protective role or has a role only in cell competition.

7. The proposed role of Rer1 in Myc cell competition needs further investigation. First, the way this manuscript is written looks like proteotoxic stress activation is associated with Myc super-competition. However, this is not the case. Proteotoxic stress activation is shown to be required for Myc overgrowth phenotype and any association with its super-competition is not known. Therefore, it is important to show that activation of proteotoxic stress activation is required for Myc super-competition.

Additionally. first paper in which the role of Myc was demonstrated in super-competition, it was also shown that differential growth is not sufficient for cell competition (de la Cova, 2004, cell). Therefore, any conclusion on Myc super-competition based on only the growth behavior of cells would lead to the wrong conclusion as shown in Fig 6 of this manuscript.

Moreover, if proteotoxic stress activation has a protective role in Myc overexpressing cells (Nagy et al 2013 PLoS Genetics), then why do these cells with simultaneous loss of Rer1 have smaller clone size? Is it possible that higher p-eIF2alpha levels shown by these cells are because of additive regulation of p-eIF2aplha levels by Myc overexpression and Rer1 loss?

8. What is the significance of Rer-1 upregulation in Myc overexpressing cells? Does Rer1 overexpression result in Myc overexpressing cells grow even better and display lower proteotoxic stress signaling.

Minor points:

1. Abstract “Cell competition is a developmental phenomenon that allows the selection of healthier cells in a developing tissue”

In the above-mentioned line in the abstract, authors have suggested cell competition as a developmental phenomenon. However, this is not the case, and it has much broader significance (Neerven et al 2022, Nature Reviews Molecular Cell Biology).

Reviewer #2: Cell Competition is key mechanism utilised to eliminate potentially dangerous cells for the living organism. In the last decade, the physiological relevance of this phenomenon has been discovered as well as players involved in the elimination of unfit cells. However, the mechanisms which induce unfit cells remain still largely unknown. Paul and Co-authors present in this article a novel Cell Competition regulator Rer1 which appears also to be required for supercompetition. The manuscript is well written and shows convincing evidence that Rer1 regulates cell fitness in wt and supercompetition. Furthermore, I believe that the manuscript will be of interest for the general audience of the journal.

However some points should be clarified or further develop:

Major points:

-Rer1 downregulation in non-competitive scenarios is sufficient to trigger cell death (Figure S1). How authors reconcile this fact with the role Rer1 in cell competition?

-Authors should test the activation of JNK pathway in competitive settings if they want to make statements about the downstream signalling (Fig.3). Given the challenging genetics, authors may want to use in this case Anti-ACTIVE® JNK to detect the activation of JNK pathway in the rer1-/- clones.

-Authors show cell competition phenotypes when downregulating or mutating rer1. It would be very interesting to analyze if Rer1 overexpression in clones is sufficient to induce supercompetitor cells in a wt background. Similarly, to test whether Rer1 overexpression in combination with Myc overexpression in clones, increases the competitive behavior of the Myc overexpressing clones alone.

Minor points:

-Figure 1B does not show the merge of RFP and DCP1, it is just showing the RFP.

-Is Figure 4A showing the experiment 96h AHS? Clones from Figure 1B-D (96h AHS) and Figure 2C (96h AHS) look much bigger.

-Fig. 6: Authors only demonstrate the requirement of Rer1 in supercompetition they do not show higher levels of Rer1 in competitive settings (i.e. Supercompetition).

-Fig. 6J, Distance is not measured in a.u.. Authors may want to say microns/pixels.

-Figure legends for some panels are missing, eg: Fig.1 E”.

-Clarify what authors refer for the sample size (N), number of discs or number of clones.

Reviewer #3: In this work Paul et al generated a rer1 null allele, which presented early larval lethality. They showed that loss of Rer1 leads to increased proteotoxic stress and JNK activation. Additionally they present evidence that Rer1 mutant clones are eliminated through cell competition. They showed that Rer1 levels are upregulated upon Myc overexpression, by using a GFP tagged rer1 genomic rescue construct. The support that Rer1 has a cytoprotective role in Myc induced proteotoxic stress, which is essential for supercompetition. Rer1 acts as a novel regulator of protein homeostasis in Drosophila and reveal its role in competitive cell survival.

Strong points: The author by generating a new Rer1 knockout allele, they manage to show that Rer1 indeed is an essential factor in Drosophila as it is in other organisms and that Rer1 mutant cells presents characteristics of loser fate. This works comes to add another example of proteotoxic driven cell competition.

Weak point: The authors aim to demonstrate that Rer1 has a protective role in Myc induced proteotoxic stress, however, the data does not fully support that the effect of Rer1 is specific to Myc induced mechanism or it is due to its essential function in the cell.

Major Revisions

1. Please provide Dcp1 staining in the experiment with the partial rescue of clonal size of rer1-/- cells by overexpression of bskDN, to show a direct effect on the cell competition hallmark, Dcp1. This is important since there is difference in the size of rer1-/- clone in Figure 2 according to time after heat shock. Therefore, any developmental delay or difference in egg deposition could have an impact in the clone size, not necessarily the competition between the clones and the background cells.

2. Could the authors show reduced Dcp1 for the experiment that they rescue clone size by overexpressing GADD34? To exclude that the clone difference is an secondary effect of developmental timing and not due to competition. (logic similar with the rescue by BskDN).

3. The authors in order to explore if Rer1 plays a role in the growth of Myc-overexpressing cells, they state that “We generated Myc-overexpressing clones in the wing disc, in either wild-type or rer1–/– background”. According to their genotype (Fig 6C : hs-FLP, UAS-GFP/+; UAS-Myc/+; tubP-Gal4, neoFRT82B, tubP-Gal80/ neoFRT 82B, ry506, rer1KO) , the background is rer1+/- heterozygous and only the myc overexpressing clones will be rer1-/-.

4. They showed in Figure 6, that the overgrowth phenotype observed due to the overexpression of Myc was reduced in the rer1–/– cells (Fig 6A – D; quantified in 6E, compare 6B and 6C), underscoring that Rer1 is required for Myc-induced overproliferation. Earlier in the manuscript authors have solid data that Rer1 is an essential protein even in wild type cells. How the authors can exclude the possibility that the absence of rer1 reduces Myc-induced overproliferation, not due to a specific effect on the Myc mechanism, but independently as an essential factor. Therefore, that would mean that Rer1 is required in the cells independently of the Myc induced mechanism.

5. What happens when the authors have the extra copy of the rer1 locus (GFP tagged). Can they see if the supercompetitor status of myc overexpressing cells is increased more with extra Rer1 protein, by checking the clone size for example, to support the cytoprotective role of Rer1 in Myc cells?

6. What happens if they have heterozygosity or homozygosity of rer1 locus in all cells. For example, would it be easy to perform the experiment using this genotype: hs-FLP, UAS-GFP/+; UAS-Myc/+; tubP-Gal4, neoFRT82B, tubP-Gal80, rer1KO / neoFRT 82B, ry506, rer1KO. In that case all the cells will lack rer1, not only the myc overexpressing clones.

7. The above additive effect could also explain the higher levels of p-eIF2a when cells overexpress Myc but lack Rer1 protein. Someone could support that the cells have two different independent stressors that increases p-eIF2a levels and reduces growth of the clones. I think it is important to strengthen this conclusion by other approaches.

Minor Revisions

1. In the abstract where they mention ER add also endoplasmic reticulum (ER)

2. Authors mention that: “Previous studies have suggested that the loser fate of Rp+/- cells is due to a reduction in protein translation [7–9]”. Actually, Lee et al 2018 (citation #7) suggested that reduced translation was likely responsible for the slow growth of Rp+/- cells, but they did not proposed that loser fate of Rp+/- cells is due to a reduction in protein translation.

3. Authors mention that: “However, recently it was shown that it is a result of increased proteotoxic stress due to protein aggregation [10–13].” The word “shown” will be misleading for the audience. I think it is better to use a word not so loud since the existed data do not clearly support that loser fate of Rp+/- cells in Drosophila is due to proteotoxic stress due to protein aggregation. There is still inconsistency in the field

4. In Figure S1 panel I, the Y axis has the “%” symbol. Do the authors mean that the ratio of [Anterior Dcp1]/ [posterior Dcp1] is 32%? Does this mean that posterior compartment has ~3 times more Dcp1 compared to anterior in hhGal4::UAS-rer1-RNAi experiment? In Figure S1F it looks quite more Dcp1 in posterior vs anterior, not just 3 fold. Also, the control hh-Gal4 in panel I should be 100, since the Dcp1 does not differ between anterior and posterior compartment. Therefore the ratio is 1 and if we are transforming this to %, must be 100%. Same comments for panel J.

5. In panel C of Figure 1, the rer1-/- clone that is located in the ventral side of the posterior compartment does not present competitive cell death at the boundaries. Contrary the twin spot (rer1+/+) shows boundary cell death. Have the author noticed any difference in rer1 clones and twin spots between ventral and dorsal compartments? Maybe they could provide more examples of clones. In this experiment (Figure 1C-D) they mention that they have generated clones 96hs after heat shock. Have the authors done Dcp1 staining in 72 hours clones, where the rer1-/- clones are bigger and in the process of elimination (according to their data in Figure 2B)?

6. In panel F of Figure 1 the authors provide the quantification of the RFP negative clones area per disc pouch. Since the clones are generated via heat shock flippase, I think it will be more appropriate to compare the ratio of RFP negative clone area compared to RFP double positive area (twin spot). They have performed this kind of ratio quantification in Figure 2A-E.

7. At what developmental stage the heat shock was done in Figure 2D, in order to have 120hrs clones (5 days). Do the larvae present a developmental delay? Could the authors provide details for the flyfood that was used for these experiments.

8. In panel Figure 2E in the Y axis the symbol (%) is misleading. The twin spot is larger than the clone therefore the percentage of the ratio must be higher than 100%. For example in 120hrs the 20% ratio that is depicted, gives the impression that twin spot is smaller than the clones.

9. For the figure 2F-I, the authors have created rer1-/- clones in RpS3+/- background. They do not present the control experiments to make rer1+/+ clones (wild type) in RpS3+/- background, to investigate if rer1-/- clones perform worsen than rer1+/+ clones when they are next to RpS3+/- cells. I do not find informative the panel 2H. It would be meaningful, if we could compare this with the control situation, where they have wild type cells next to RpS3+/-. In that case they could compare if Dcp1 in RpS3+/- cells is different when the cells next to them have wild type rer1 (rer1+/+) or rer1-/- mutant. This experiment also shows that rer1-/- cells can still outcompete RpS3+/- cells.

10. Have the authors tested if the heterozygosity of the null rer1 allele has any impact in Minutes. In the above experiments, the Minutes have also lost one copy of rer1 gene.

11. In legend 2F they mention”hs-FLP-induced heterozygous RpS3+/– clones (GFP-positive cells) juxtaposed to rer1–/– cells (GFP-negative), and immuno-stained for the anti-cleaved Dcp-1.”. According to their genotype, using the flippase they create rer1-/- clones in RpS3+/- background (the twin spot of these clones were RpS3-/-, rer1+/+, but the cells do not survive since RpS3 is an essential protein). Therefore, the most accurate description is ” hs-FLP-induced rer1–/–clones (GFP-negative) juxtaposed to heterozygous RpS3+/– cells (GFP-positive cells), and immuno-stained for the anti-cleaved Dcp-1.” They use also the term “RpS3+/- clones” in the text. It should be RpS3+/- background

12. Comment for Figure 4A: Could the authors provide another disc with rer1-/- clones, where they will also depict p-eIF2a levels in comparison with DAPI staining (since p-eIF2a levels are cytoplasmic)? For example, in Panel 4B’’ we can see higher levels of p-eIF2a in the top-edge of the Figure where there is no rer1-/- cells. Maybe with the 72 hours AHS mitotic clones, which are bigger in size, the increase in p-eIF2a in rer1-/- mutant clones is more prominent. Indeed, in Figure 5B the increased levels of p-eIF2a in rer1-/- clones is more prominent than in 4A.

13. In Figure S5 could they correlate the DHE staining with the nuclei position. In panel B’’’ there is an area in the anterior compartment with increased DHE, which actually is mainly cytoplasmic, which could be responsible for this. I kind of agree that it seems that DHE is increased in posterior compartment, but I would like to compare different areas of the two compartments with the same nuclei positions.

Thank you.

**Have all data underlying the figures and results presented in the manuscript been provided?**

Reviewer #1: Yes

Reviewer #2: Yes

Reviewer #3: None

PLOS authors have the option to publish the peer review history of their article (what does this mean?). If published, this will include your full peer review and any attached files.

Reviewer #1: No

Reviewer #2: **Yes: **Marisa M.Merino

Reviewer #3: **Yes: **Marianthi Kiparaki

---

## [Decision Letter · Decision Letter 1]

5 Jan 2024

Dear Dr Chaudhary,

Thank you very much for submitting your Research Article entitled 'Maintenance of proteostasis by Drosophila Rer1 is essential for competitive cell survival and Myc-driven overgrowth' to PLOS Genetics.

The manuscript was fully evaluated at the editorial level and by independent peer reviewers. The reviewers and editors appreciate the effort in providing additional data to address the reviewers' concerns.  While these revisions were not sufficient to sway reviewer 1, the editors agree with the viewpoint of reviewers 2 and 3.  However, reviewer 3 raises some minor points about Fig. 9 that need to be addressed before the manuscript can be accepted for publication. 

We therefore ask you to modify the manuscript according to reviewer 3's recommendations. Your revisions should address the specific points made by reviewer 3.

Yours sincerely,

Ken M. Cadigan, PhD

Academic Editor

PLOS Genetics

Gregory P. Copenhaver

Editor-in-Chief

PLOS Genetics

Reviewer's Responses to Questions

**Comments to the Authors:**

Reviewer #1: The manuscript looks much better than last time. However, revisiting my previous comments, a major criticism was the absence of an in-depth study. I still feel this aspect has not been adequately addressed. The mechanism governing Rer1 maintenance protein homeostasis remains unclear. As I inquired before: Does Rer1 play a direct or indirect role in activating Perk, and how it contributes to protein homeostasis needs to be thoroughly examined.

Reviewer #2: The authors have now clarified all my concerns and the new version of the manuscript shows further conceptual and experimental insights of this new Cell Competition player.

I recommend the manuscript for publication in PLOS Genetics.

Reviewer #3: Dear Paul, Umarvaish et al,

I find the responses to the reviewer's questions detailed and satisfactory. Thank you.

I have some minor concerns regarding the role of Rer1 in Myc supercompetition.

Minor concerns:

A) Could the authors explain why they were unable to perform the experiment to analyze Myc-driven overgrowth in flies with an extra copy of the rer1 gene (GFP-rer1)? Did the larvae with the appropriate genotype (AFG/hs-FLP; UAS-Myc/GFP-rer1; UAS-GFP/+) die? Were they unable to retrieve clones? I find important for the field to present (or at least mention) the result of Rer1 overexpression.

B) Please provide some details and clarifications of the experiment that is provided in the Figure 9 and gives better support on the role of Rer1 in Myc supercompetition.

More specifically:

1) Could the authors provide the details on the heat shock time and dissection time in the experiment Fig. 9A-B.

2) Is the p-eIF2a ratio in Fig 9C increased in AFG::Myc, GFP, rer1+/- experiment compared to AFG::Myc, GFP due to the decrease of p-eIF2a in rer1+/- background cells? Is the ratio increased because the the Myc, rer1+/- cells have increased p-eIF2a? The images provided in the pdf version are not clear and they give the impression that the p-eIF2a staining is similar in AFG::Myc cells and in AFG::Myc, rer1+/- cells.

3)Have the authors performed control clones in WT and in rer1+/- background, meaning AFG without over-expressing Myc. Do the clones look smaller in rer1+/- background?

4) If the authors have discs that the Myc, GFP cells have not undertaken the whole disc would be useful. Also show discs with similar size (the 9A disc looks bigger than 9B).

C) Please mention that also in Ochi et al, PERK depletion suppressed the elimination of wol clones (wol is involved in the glycosylation of proteins in the ER), of Hel25E clones.

Best wishes,

Marianthi

**Have all data underlying the figures and results presented in the manuscript been provided?**

Reviewer #1: Yes

Reviewer #2: Yes

Reviewer #3: Yes

PLOS authors have the option to publish the peer review history of their article (what does this mean?). If published, this will include your full peer review and any attached files.

Reviewer #1: No

Reviewer #2: **Yes: **Marisa M.Merino

Reviewer #3: **Yes: **Marianthi Kiparaki

---

## [Decision Letter · Decision Letter 2]

5 Feb 2024

Dear Dr Chaudhary,

We are pleased to inform you that your manuscript entitled "Maintenance of proteostasis by Drosophila Rer1 is essential for competitive cell survival and Myc-driven overgrowth" has been editorially accepted for publication in PLOS Genetics. Congratulations!

Yours sincerely,

Ken M. Cadigan, PhD

Academic Editor

PLOS Genetics

Gregory P. Copenhaver

Editor-in-Chief

PLOS Genetics

Comments from the reviewers (if applicable):

Reviewer's Responses to Questions

**Comments to the Authors:**

Reviewer #3: Dear authors,

Thank you for your revised version.

I am completely satisfied with your replies to my previous comments.

**Have all data underlying the figures and results presented in the manuscript been provided?**

Reviewer #3: Yes

PLOS authors have the option to publish the peer review history of their article (what does this mean?). If published, this will include your full peer review and any attached files.

Reviewer #3: **Yes: **Marianthi Kiparaki

**Data Deposition**

http://datadryad.org/submit?journalID=pgenetics&manu=PGENETICS-D-23-00557R2

**Press Queries**

---

## [Editor Report · Acceptance letter]

20 Feb 2024

PGENETICS-D-23-00557R2 

Maintenance of proteostasis by Drosophila Rer1 is essential for competitive cell survival and Myc-driven overgrowth 

Dear Dr Chaudhary, 

We are pleased to inform you that your manuscript entitled "Maintenance of proteostasis by Drosophila Rer1 is essential for competitive cell survival and Myc-driven overgrowth" has been formally accepted for publication in PLOS Genetics! Your manuscript is now with our production department and you will be notified of the publication date in due course.

With kind regards,

Zsofi Zombor

PLOS Genetics

On behalf of:
